# EvoLM: In Search of Lost Training Dynamics for Language Model Reasoning

**Zhenting Qi**[1]    **Fan Nie**[2]    **Alexandre Alahi**[3]    **James Zou**[2]
**Himabindu Lakkaraju**[1]    **Yilun Du**[1]    **Eric Xing**[4]    **Sham Kakade**[1]    **Hanlin Zhang**[1]
[1]Harvard    [2]Stanford    [3]EPFL    [4]CMU

## Abstract

Modern language model (LM) training has been divided into multiple stages, making it difficult for downstream developers to evaluate the impact of design choices made at each stage. We present **EvoLM**, a model suite that enables systematic and transparent analysis of LMs' training dynamics across pre-training, continued pre-training, supervised fine-tuning, and reinforcement learning. We train over 100 LMs with 1B and 4B parameters from scratch, and evaluate both upstream (language modeling) and downstream (problem-solving) capabilities, including considerations of both in-domain and out-of-domain generalization. Key insights highlight the diminishing returns from excessive pre-training and post-training, the importance and practices of mitigating forgetting during domain-specific continued pre-training, the crucial role of continued pre-training in bridging pre-training and post-training phases, and various intricate trade-offs when configuring supervised fine-tuning and reinforcement learning. To facilitate open research and reproducibility, we release all pre-trained and post-trained models, training datasets for all stages, and our entire training and evaluation pipeline.

🔗 **Model Suite**    🤗 **Datasets**    ⭕ **Code**

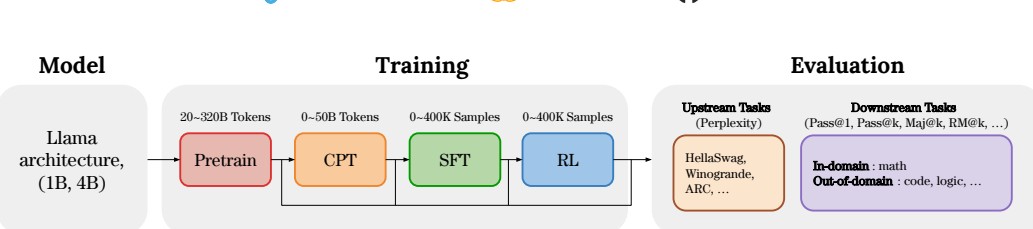

Figure 1: Overview of EvoLM, a transparent model suite for studying language-model training dynamics across pre-training, continued pre-training (CPT), supervised fine-tuning (SFT), and reinforcement learning (RL). The framework evaluates both upstream (language modeling) and downstream (problem-solving) performance across in-domain (e.g., math) and out-of-domain (e.g., code, logic) settings, enabling systematic analysis of design trade-offs and scaling behaviors.

## 1 Introduction

Scaling up language models has been a paradigm that enables various downstream applications [8, 52, 32]. One approach to understanding scaling—and enabling more efficient resource allocation—is through scaling laws, which characterize the quantitative relationship between pre-training log-loss and compute [22, 27, 23, 21]. In part due to the vast design space [34] and the complex interactions of several training phases such as pre-training and post-training [13, 70] for open-weight models [17], it remains challenging to clearly identify which decisions consistently lead to reliable downstream performance gains.

39th Conference on Neural Information Processing Systems (NeurIPS 2025).

Although progress has been made in understanding how models learn during training [59, 53, 43, 16, 66], accurately forecasting downstream problem-solving performance remains challenging due to the training-inference mismatch in auto-regressive generative models [46] and the non-smooth nature of downstream performance improvements [45]. Existing studies often rely on checkpoints with limited transparency regarding training details, which can introduce potential confounding factors, including (1) dependence on opaque analyses from post-training studies that utilize off-the-shelf base models, often without strict control over key variables such as model size, pre-training data size, and data components [42, 10, 61], and (2) evaluations based on intermediate checkpoints [59, 51], which may have sub-optimal downstream performance due to incomplete learning rate decay [48, 24, 54, 67, 30], thereby complicating fair comparisons.

In this work, we establish an end-to-end development pipeline using open toolkits [1, 71, 49] and open data sources [38, 63, 55, 31] to systematically and transparently investigate language models' reasoning capabilities throughout their lifecycle, covering phases of pretraining, continued pretraining, supervised fine-tuning, and reinforcement learning. We introduce **EvoLM**, a model suite comprising 100+ decoder-only autoregressive LMs with 1B and 4B parameters, each trained from scratch with complete learning rate decay across various configurations of model size and dataset scale. Pre-trained on publicly available corpora FineWeb [38] only, our base models achieve competitive performance on English-only language modeling tasks compared with other open-weight models with significantly more pretraining compute (Table 4). For example, our 1B and 4B models, pre-trained on 320B tokens, perform competitively with TinyLlama-1B and Qwen1.5-4B, respectively, despite their significantly more pre-training data (2T and 3T tokens). We evaluate both upstream language modeling performance (measured by perplexity) and downstream practical problem-solving capabilities (assessed through generative rollout performance) on both in-domain (ID) math reasoning and out-of-domain (OOD) general reasoning tasks. Through extensive controlled and transparent experiments, our study addresses several critical gaps in understanding LM training dynamics, provides insights into model behaviors, and identifies open research directions in recent literature. In summary, our contributions include:

- Systematic analyses of language model capabilities across their entire lifecycle—from pre-training to RL post-training—with evaluation on reasoning-intensive upstream cloze tasks and downstream generative tasks, considering both in-domain and out-of-domain generalization.

- Open-sourcing 100+ LMs trained from scratch with 1B and 4B parameters and their training data for all stages, enabling the research community to build upon our findings.

- Open-sourcing a comprehensive, transparent, and reproducible training pipeline and evaluation framework, facilitating further research into scaling laws, training dynamics, and evaluating upstream and downstream capabilities of language models.

## 2 Experimental Settings

### 2.1 Training Setup

We initialize all models using the LLaMA-2 [56] architecture with 1B and 4B parameters. Our training pipeline consists of four sequential stages:

- **Pre-training**: Conducted on FineWeb-Edu [38]. Guided by the Chinchilla scaling law [23] that recommends a compute-optimal ratio of approximately 20 tokens per model parameter, we pre-train models across token budgets ranging from the optimal 20x model size to 320B tokens to investigate the effects of mild over-training (>1x Chinchilla, ≤16x Chinchilla) and excessive over-training (>16x Chinchilla) on task performance.

- **Continued Pre-training (CPT)**: Performed on FineMath [2] with token budgets from 2B to 42B. To mitigate catastrophic forgetting of general-domain knowledge, we also incorporate pre-training data replay strategies [25, 41, 4, 62].

- **Supervised Fine-Tuning (SFT)**: Applied to a dataset of QA pairs augmented from GSM8K [12] and MATH [20], collected from a mixture of MetaMathQA [63], OpenMathInstruct2 [55], and NuminaMath [31]. We filter out low-quality prompts using model correctness consistency [39], discarding samples with zero inter-model consensus.

- **Reinforcement Learning (RL):** Conducted using Proximal Policy Optimization (PPO) [47], with a binary verifiable reward. The RL stage uses the same data sources as SFT but ensures no overlap with the SFT dataset.

We use a compact model signature to denote the configuration of each model across training stages. For example, 1B-160BT-8+42BT-100Kep1-100Kep16 represents a model with the following setup:

- 1B: A model with 1 billion parameters.
- 160BT: Pretrained on 160 billion tokens from FineWeb-Edu.
- 8+42BT: Continued pretrained with 8 billion tokens of replayed general-domain data (FineWeb-Edu) and 42 billion tokens of domain-specific data (FineMath).
- 100Kep1: Supervised fine-tuned on 100K examples for 1 epoch.
- 100Kep16: Reinforcement learning fine-tuned on 100K examples for 16 epochs.

For all configurations, we train models with complete learning rate scheduling and only take the final checkpoints as subjects of study. More training details can be found at Section C.2.

## 2.2 Evaluation Protocol

**Upstream Cloze Tasks**    These tasks assess models' language modeling capabilities via next-token prediction, without requiring conversational abilities. We evaluate pretrained and continued-pretrained models on the following datasets, reporting average 0-shot accuracy across them: HellaSwag [65], Winogrande [44], PIQA [6], OBQA [36], ARC-Easy/Challenge [11].

**Downstream Generative Tasks**    These tasks evaluate models' problem-solving abilities in a generative, conversational setting. We test supervised fine-tuned and RL-finetuned models on: 1) *In-Domain Tasks (math reasoning):* GSM8K-Platinum [57] (a revised version of the full GSM8K [12] test set that minimizes label noises) and MATH [20]. 2) *Out-of-Domain Tasks:* CRUXEval [19] (code reasoning), BGQA [28] (logical reasoning), TabMWP [35] (table reasoning), and StrategyQA [18] (commonsense reasoning). We evaluate models in a zero-shot manner by prompting them to generate full solutions in response to problems and report average performance for ID and OOD tasks. More evaluation details including dataset descriptions, sampling parameters, and standard errors are reported in Section C.3. Evaluation metrics include:

- **Accuracy:** We measure accuracy under four prompting schemes: 1) **Pass@1:** Temperature = 0. A single deterministic response is generated. The problem is marked correct if this response is correct. 2) **Maj@16:** Temperature = 1. Sixteen responses are sampled, and the majority answer is evaluated for correctness. 3) **RM@16:** Temperature = 1. Sixteen responses are sampled; the one with the highest ORM score is evaluated for correctness. 4) **Pass@16:** Temperature = 1. Sixteen responses are sampled; the problem is marked solved if any one of the responses is correct.

  For all these settings, final answers are extracted from model outputs and compared against ground-truth solutions to determine correctness. We additionally report **Correct Ratio**: In the response groups that have at least one correct solution, we compute the ratio of the number of correct solutions to the total number of solutions (16).

- **ORM Score:** We use an outcome reward model—`Skywork-Reward-Llama-3.1-8B-v0.2` [33]—to assign scalar scores to generated solutions, based on input problems and responses. This metric serves as a proxy for solution quality.

# 3 Scaling Studies Across Three Training Stages

## 3.1 Scaling Up Pre-training Compute

To quantify how varying the total amount of pre-training compute affects language modeling performance, we pre-train 0.5B, 1B, 4B models on token budgets ranging from 10 B up to 320 B tokens. As shown in Figure 2, performance on upstream tasks improves steadily with more pre-training tokens, but with rapidly diminishing returns beyond around 80x to 160x model size. For example, the 1B model's average accuracy increases from roughly 46% at 20 BT to 52% at 80 BT, yet gains shrink to

less than a percentage point when moving from 80 BT to 160 BT. The larger 4B model continues to benefit slightly longer but also plateaus by 320 BT.

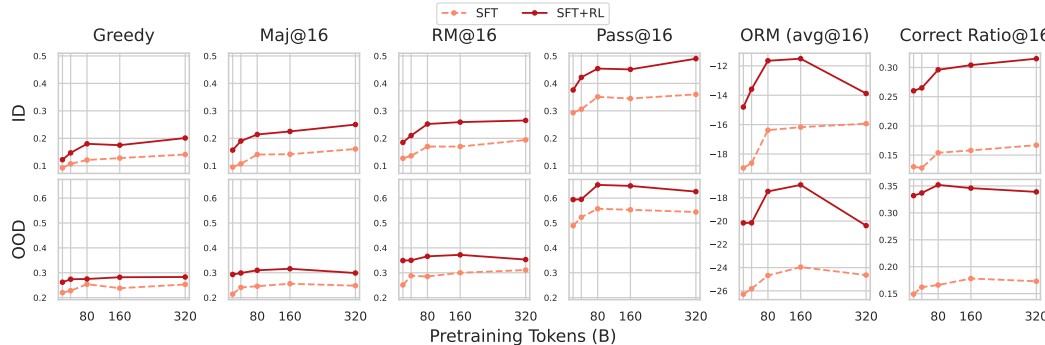

Figure 3: Downstream task performance vs. number of pretraining tokens on models:
- **SFT**: 1B-{20BT, 40BT, 80BT, 160BT, 320BT}-8+42BT-100Kep1
- **SFT+RL**: 1B-{20BT, 40BT, 80BT, 160BT, 320BT}-8+42BT-100Kep1-100Kep8.

We further assess how these pre-training budgets translate to downstream capabilities for both SFT and SFT+RL models. Figure 3 shows all six metrics on ID and OOD downstream tasks from 20BT to 320BT pretraining budgets for 1B models. Both SFT and SFT+RL variants exhibit strong initial gains up to 80BT, but performance saturates thereafter: For instance, ID Maj@16 accuracy of SFT model rises sharply from 8% at 20 BT to 15% at 80 BT, yet only inches up to 17% at 320 BT. RL yields a consistent uplift over pure SFT, but likewise shows negligible benefit from over-training beyond 80BT. Moreover, Maj@16, RM@16, and Pass@16 accuracies on OOD tasks decrease after 160BT budget, and such degradation is also amplified by a drop in ORM score,

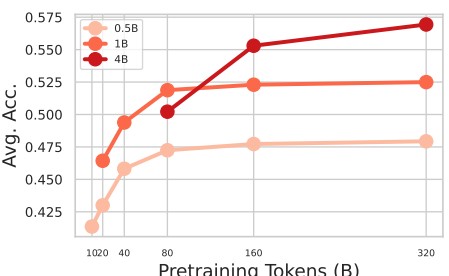

Figure 2: Upstream task performance vs. pre-training tokens on models {0.5B, 1B, 4B}-{10BT, 20BT, 40BT, 80BT, 160BT, 320BT}.

showing the overall generation quality decreases to a certain amount. These patterns reveal that excessively large pre-training budgets also lead to diminishing returns on downstream performance and might even cause degradation. This finding is consistent with previous work [51], which points out that scaling up pre-training does not always improve or can even hurt LMs' performance after SFT, and we further complete the studies by showing that 1) such performance gain stagnation is also reflected on downstream generative reasoning tasks and 2) RL finetuning is also constrained by overtraining.

➟ **Takeaway 1.** *Excessive general-domain pre-training does not always improve domain-specific post-training and might even cause performance degradation on some downstream tasks (saturation happens around 80x to 160x model size in our study).*

We further look into how model size interplays with scaling up pre-training. As Table 1 illustrates, under a fixed pre-training compute budget (1B–320BT vs. 4B–80BT), the smaller 1B model even outperforms the 4B model across both SFT and SFT+RL settings. When matching on pre-training tokens, we see the same trend at lower budgets: at 80B tokens the 1B–80BT and 4B–80BT models perform comparably, with the smaller model slightly ahead. However, once the budget rises to 160B tokens, the 4B–160BT model "unlocks" its scale: For example, the 4B SFT model jumps to an ID Maj@16 of 26.4% (vs. 14.2% of 1B counterpart) and the 4B SFT+RL model jumps to 34.8% (vs. 22.5% of 1B counterpart), demonstrating that only after reaching the saturation regime of pre-training does model size translate into substantial gains in post-training performance.

➟ **Takeaway 2.** *Under limited pre-training budgets, smaller post-trained models can even outperform larger counterparts. Conversely, once pre-training tokens reach the saturation regime, increasing model size enables clear improvements in both in-domain performance and OOD generalization.*

| Base Model | ID Acc. (SFT / SFT+RL) | | | OOD Acc. (SFT / SFT+RL) | | |
|---|---|---|---|---|---|---|
| | **Greedy** | **Maj@16** | **Pass@16** | **Greedy** | **Maj@16** | **Pass@16** |
| **Same Pretraining Compute** | | | | | | |
| 1B-320BT-8+42BT | 14.1 / 20.1 | 16.1 / 25.0 | 36.0 / 49.0 | 25.3 / 28.3 | 24.8 / 29.9 | 54.4 / 62.6 |
| 4B-80BT-8+42BT | 11.3 / 15.7 | 13.2 / 20.0 | 34.2 / 43.0 | 24.8 / 28.2 | 23.4 / 29.6 | 52.2 / 60.2 |
| **Same Pretraining Tokens** | | | | | | |
| 1B-80BT-8+42BT | 12.1 / 18.0 | 14.1 / 21.4 | 35.1 / 45.4 | 25.4 / 27.5 | 24.6 / 31.0 | 55.7 / 65.3 |
| 4B-80BT-8+42BT | 11.3 / 15.7 | 13.2 / 20.0 | 34.2 / 43.0 | 24.8 / 28.2 | 23.4 / 29.6 | 52.2 / 60.2 |
| 1B-160BT-8+42BT | 12.8 / 17.5 | 14.2 / 22.5 | 34.5 / 45.1 | 23.8 / 28.2 | 25.6 / 31.6 | 55.3 / 64.9 |
| 4B-160BT-8+42BT | 22.0 / 27.8 | 26.4 / 34.8 | 47.6 / 58.4 | 27.9 / 29.6 | 26.0 / 33.2 | 57.3 / 66.2 |

Table 1: Comparison between 1B and 4B **SFT / SFT+RL** models under fixed pre-training compute/-tokens.

## 3.2 Scaling Up Continued Pre-training Compute

We investigate the impact of continued pretraining (CPT) compute by varying the total CPT tokens from 0 (no CPT) to 50 BT, using 1B-160BT pretrained model as the base. As shown in Figure 4, increasing CPT compute gradually degrades upstream task performance, indicating *catastrophic forgetting* [15]. To mitigate this issue, we adopt a simple "replay" strategy [25] by randomly interleaving a small amount of pre-training data during CPT. Figure 4 demonstrates that the model with 8 BT replay consistently maintains higher upstream accuracy than the no-replay baseline across all CPT budgets. We then apply SFT on the CPT models on 100K exam-

| CPT Config | Acc. |
|---|---|
| No CPT | 6.04 |
| FineMath 50BT | 19.27 |
| FineWeb 1.6BT + FineMath 48.4BT | 16.21 |
| FineWeb 8BT + FineMath 42BT | **21.01** |
| FineWeb 16BT + FineMath 34BT | 15.22 |

Table 2: GSM8K-Platinum performance (Pass@1 accuracy) of pretrained model 1B-160BT continued pretrained with various configurations and then finetuned using 100K SFT examples with 1 epoch.

ples for one epoch to investigate the impact of replay on downstream performance. Table 2 reports Pass@1 accuracy on GSM8K-Platinum for each CPT mix. Pure FineMath CPT (50 BT) achieves 19.27%, whereas a mix of 8 BT FineWeb replay with 42 BT FineMath tokens even yields a better result at 21.01%. Configurations with either too little (1.6+48.4 BT) or too much (16+34 BT) replay perform worse, highlighting that a modest replay budget (around 5%) optimally balances retention of general-domain knowledge with adaptation to downstream generative tasks.

➠ **Takeaway 3.** *CPT on domain-specific data induces catastrophic forgetting of pre-trained knowledge which could harm both upstream and downstream performance, while incorporating a small replay budget (e.g. 5%) could effectively mitigate this degradation.*

In Figure 5, we plot downstream performance of both SFT and SFT+RL models as a function of CPT budget (with a fixed 8 BT replay budget). All variants improve steadily with more domain-specific tokens up to around 32 BT and then plateau by 42 BT. For instance, the ID greedy accuracy of the SFT model rises from about 5% at 2 BT to 12% at 32 BT before leveling off. Such a trend is also observed in OOD metrics. Across the CPT range, RL finetuning consistently outperforms pure SFT; notably, without CPT, RL can even underperform SFT (as seen

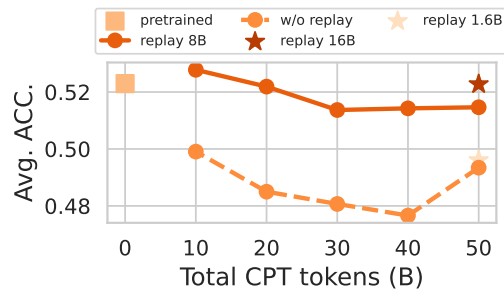

Figure 4: Upstream task performance vs. CPT tokens on models:
- **Pretrained:** 1B-160BT,
- **CPT:** 1B-160BT-8+{2BT, ..., 42BT},
- **CPT:** 1B-160BT-0+{10BT, ..., 50BT},
- **CPT:** 1B-160BT-{1.6+48.4BT, 16+34BT}.

in Maj@16, RM@16, and Pass@16), yet the gain brought by RL tends to strengthen as CPT tokens increase.

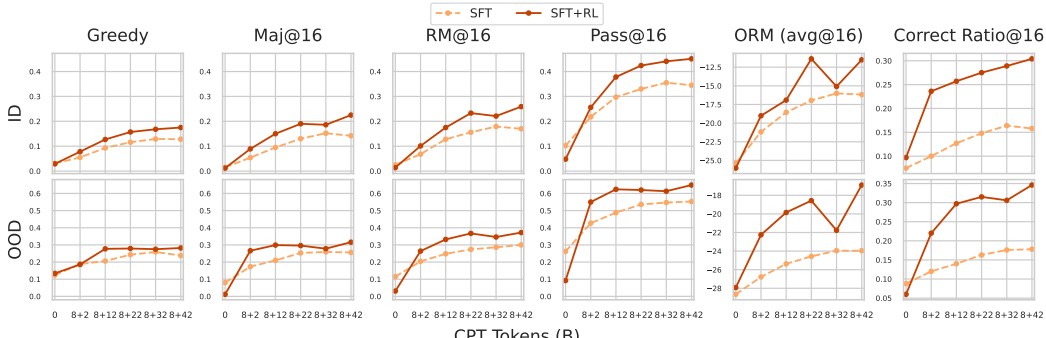

Figure 5: Downstream task performance vs. continued pre-training tokens on models:
- **SFT:** 1B-160BT-100Kep1, 1B-160BT-8+{2BT, ..., 42BT}-100Kep1
- **SFT+RL:** 1B-160BT-100Kep1-100Kep8, 1B-160BT-8+{2BT, ..., 42BT}-100Kep1-100Kep8.

➠ **Takeaway 4.** *Domain-specific post-training should be supported by adequate domain-specific CPT data: without it, SFT performance remains suboptimal and RL can even degrade such performance.*

➠ **Takeaway 5.** *As domain-specific CPT data increase, in-domain downstream performance steadily improves and the SFT models could benefit more from RL finetuning.*

➠ **Takeaway 6.** *With sufficient domain-specific CPT data, post-training on in-domain tasks not only improves in-domain performance but also generalizes effectively to OOD tasks.*

## 3.3 Scaling Up SFT Compute

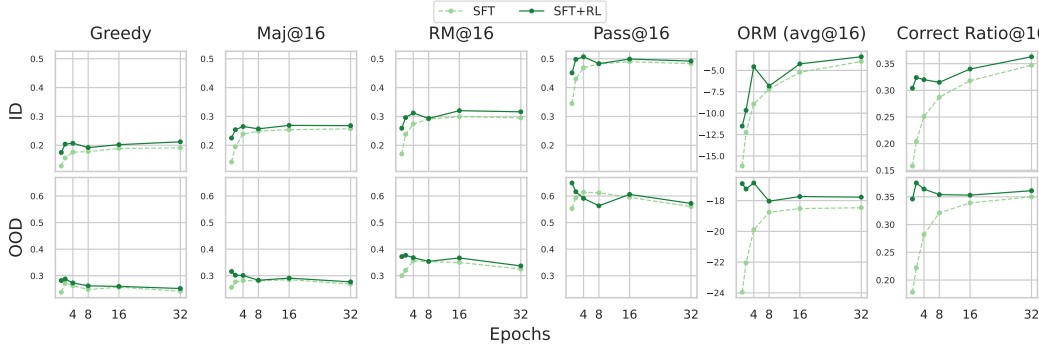

Figure 6: Downstream task performance vs. number of SFT epochs for models:
- **SFT:** 1B-160BT-8+42BT-100Kep{1,2,4,8,16,32}
- **SFT+RL:** 1B-160BT-8+42BT-100Kep{1,2,4,8,16,32}-100Kep8.

To evaluate how downstream performance responds to increased SFT compute, we conduct two complementary studies using 1B-160BT-8+42BT as the base model.

**Varying SFT epochs.** Holding SFT examples fixed at 100K, we finetune the base model for {1, 2, 4, 8, 16, 32} epochs. As shown in Figure 6, ID metrics increase steadily with more epochs and saturate at around 8 epochs, reflecting increased memorization of solving in-domain problems. In contrast, OOD performance peaks at 2–4 epochs before declining, indicating that over-specialization hinders generalization. These findings also validate the commonly chosen SFT hyperparameter of approximately 3 epochs. Moreover, the marginal gains from downstream RL finetuning shrink on over-trained SFT models: once the model has excessively memorized the supervised data, there is little room for RL to improve.

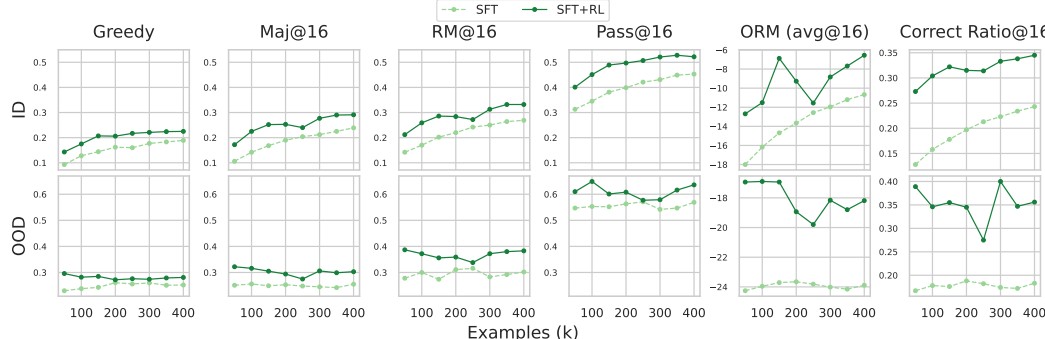

Figure 7: Downstream task performance vs. number of SFT examples for models:
- **SFT:** 1B-160BT-8+42BT-{50K, 100K, 150K, ..., 400K}ep1
- **SFT+RL:** 1B-160BT-8+42BT-{50K, 100K, 150K, ..., 400K}ep1-100Kep8.

**Varying SFT dataset size.** As proposed by previous study [42], post-training performance for downstream tasks follows a power-law relationship with SFT dataset size, but the conclusion is drawn from experiments conducted on up to 10K examples. We further scale that budget by varying the number of SFT examples from 50K to 400K, holding epochs fixed at one to minimize memorization. As illustrated in Figure 7, ID performance improves monotonically with more examples, confirming that additional SFT compute consistently improves performance on in-domain tasks. However, OOD metrics fluctuate and can even decline with larger datasets. Similarly as scaling up epochs, the incremental benefit from RL diminishes as the model learns more SFT examples.

➥ **Takeaway 7.** *Excessive SFT improves ID performance with diminishing returns but does not necessarily improve and can even degrade OOD performance.*

➥ **Takeaway 8.** *Excessive SFT, especially overly large epochs, could limit further RL improvements.*

### 3.4 Scaling Up RL Compute

Similarly, to evaluate how downstream performance responds to increased RL compute, we also vary either epochs or dataset size, using a 1B-160BT-8+42BT-100Kep1 base model. We additionally incorporate 0 epochs/examples to indicate the SFT baseline. More experiment results and findings regarding RL can be found at Section B.2.

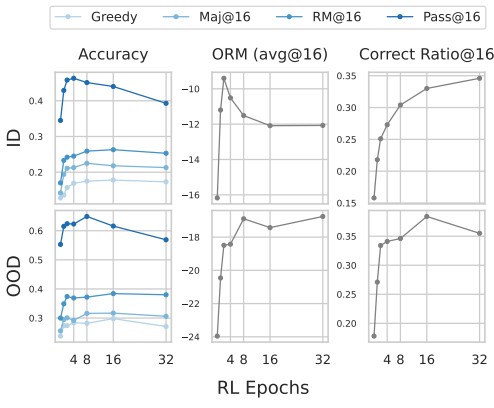

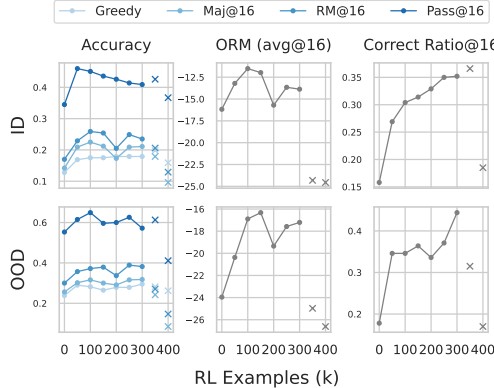

(a) Performance v.s. number of RL epochs for models 1B-160BT-8+42BT-100Kep1-100Kep{0, 1, 2, 4, 8, 16, 32}.

(b) Performance v.s. number of RL examples for models 1B-160BT-8+42BT-100Kep1-{0, 50K, 100K, 150K, ..., 400K}ep1.

Figure 8: Downstream task performance under different RL scales.

**Varying RL epochs.**   We apply RL across another 100K examples (disjoint from the SFT dataset) for {0, 1, 2, 4, 8, 16, 32} epochs. As shown in Figure 8a, for both ID and OOD tasks, greedy, Maj@16, and RM@16 accuracies peak at around 8–16 epochs and then saturates thereafter. We also notice that while the correct ratio keeps increasing, Pass@16 accuracy greatly degrades beyond 4 epochs, indicating that RL primarily sharpens confidence in already-correct outputs rather than effectively expanding the set of solvable samples. This is also reflected by results in Table 1: For 1B and 4B SFT models, Maj@16 accuracy could sometimes underperform greedy accuracy, indicating that low-quality solutions take the majority. However, after RL is conducted on the SFT models, all Maj@16 accuracies are higher than greedy accuracies.

**Varying RL dataset size.**   Given a fixed epoch of 8, we vary the RL dataset size from 0 to 400K examples. Figure 8b shows that for both ID and OOD metrics, greedy, Maj@16, and RM@16 accuracies continue to increase from more data up to around 150–200K examples, after which gains flatten and fluctuate. In contrast, Pass@K saturates much earlier and starts to degrade, while the correct ratio keeps increasing, similar to the finding in scaling up RL epochs. This finding is in line with observations by concurrent work [64] that similarly conclude that RL mainly boosts the confidence of existing correct outputs rather than enhancing the fundamental reasoning capabilities of LMs. We further expand this insight by illustrating the precise trade-offs for both RL epochs and dataset size. Additionally, we notice a drastic performance drop at 350K and 400K examples, and training results show that during the final RL steps, both models learn to greatly increase response length and their generations often exceed their predefined context window lengths, thus causing the performance drop. However, RL with overly large epochs is much more stable and such collapse caused by response length scaling is not observed.

⇛ **Takeaway 9.** *RL with excessive epochs or examples improves downstream performance on both ID and OOD tasks but with diminishing returns (saturation happens at 4-8 epochs or 50-100K examples in our study).*

⇛ **Takeaway 10.** *Beyond saturation regime, RL primarily increases the probability of sampling high-quality rollouts but does not necessarily improve models' fundamental reasoning capabilities.*

To further investigate how to configure SFT and RL data allocation in data-constrained scenarios, we subsample 100K examples from the entire 500 K dataset and evaluate five SFT/RL splits: (10 / 90, 30 / 70, 50 / 50, 70 / 30, 90 / 10) K and conduct either SFT or RL for 4 epochs. We choose 100K because it is around the saturation regime of both ID and OOD performance (Figure 8b). As shown in Figure 9, ID accuracy (greedy and Pass@16) increases with the proportion of SFT data, plateauing beyond around 70 K, whereas OOD metrics are driven by RL allocation, peaking at 10K SFT (i.e. 90K RL). These trends hold across both the 1B and 4B models.

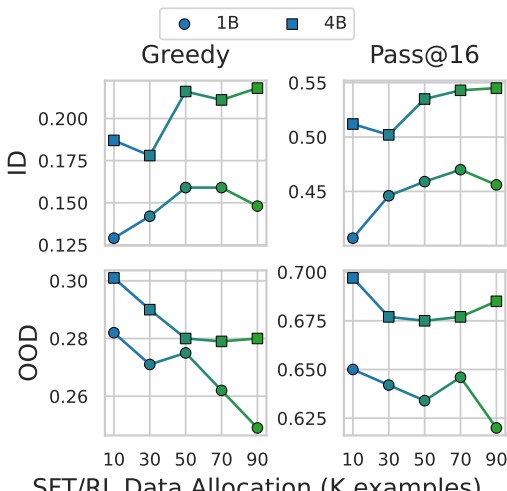

Figure 9: Downstream task performance for {1B, 4B}-160BT-8+42BT-{10K, ..., 90K}ep4-{90K, ..., 10K}ep4. Darker green/blue denotes more data allocation to SFT/RL. The total number of post-training samples is fixed at 100K.

⇛ **Takeaway 11.** *Under a constrained downstream data budget, allocating more examples to SFT maximizes in-domain gains at the expense of weaker OOD generalization, while allocating more to RL improves OOD performance.*

## 4   Additional Studies and Discussions

Given that we find post-training interacts non-trivially with pre-training—necessitating a sophisticated training recipe—does downstream performance scale smoothly or predictably? This section provides one example illustrating why our comprehensive study is essential to fully grasp how training

dynamics shape downstream performance in LMs, and another example where a metric could correlate with downstream problem-solving performance.

## 4.1 Intermediate Checkpoints May Not Be Reliable Surrogates

In reality, practitioners usually train each desired model through the full learning-rate schedule and exhaust the available pre-training data, rather than taking intermediate checkpoints as final models. To mimic the real-world workflow of training models from scratch for 20B or 40B tokens, we compare those standalone runs against the checkpoints extracted at the same token counts (20B and 40B) from a longer 160B-token pre-training run. After each model sees 20B or 40B tokens, we further apply a single epoch of SFT on 100K examples to deliver a basic conversational grounding, and evaluate the models on two easiest subsets of the MATH dataset.

| Model | Upstream | Downstream (Greedy / Pass@16) | |
| | | Math Level 1 | Math Level 2 |
| --- | --- | --- | --- |
| 20BT full | 46.43 | 2.75 / 17.85 | 3.36 / 15.10 |
| 20BT int. | 46.07 | 2.52 / 11.44 | 1.90 / 12.64 |
| 40BT full | 49.38 | 2.97 / 17.96 | 3.36 / 14.88 |
| 40BT int. | 49.06 | 1.37 / 9.38 | 2.68 / 8.72 |

Table 3: Performance on Upstream tasks and MATH (Level 1 and 2) under different pretraining configurations. "$x$BT full" refers to a complete pre-training run on $x$ BT, while "$x$BT int." refers to an intermediate checkpoint taken during training to 160B tokens, corresponding to $x$ BT seen so far.

As Table 3 shows, the intermediate checkpoints consistently lag behind their dedicated 20B and 40B counterparts on both upstream task accuracy and math reasoning performance. This gap arises because earlier stopping points–captured before learning-rate decay and data repetition–omit the full optimization trajectory that smaller runs complete. In other words, simply slicing out a 40B-token checkpoint from a longer schedule does not reproduce the benefits of training a model exclusively for 40B tokens.

These results caution against using such intermediate checkpoints as proxies for studying and understanding fully trained smaller models. When interpreting training dynamics, it is essential to compare like-for-like runs—each with its own complete schedule—rather than relying on mid-course snapshots that understate true model capability.

## 4.2 Correlating Downstream Task Performance with ORM Score

While perplexity across domains sometimes shows strong correlations, downstream task accuracy may not be consistently correlated, largely because post-trained models are miscalibrated and thus lower validation perplexity does not necessarily indicate better generative performance. In our experiments, we found that the correlation between ORM scores and downstream task accuracy presents a clear relationship. In Figure 10, we plot ORM score (avg@16) versus Maj@16 accuracies for all post-trained model variants starting from base model 1B-160BT-8+42BT and find that ORM scores exhibit consistently strong predictive power, evidenced by high correlation coefficients ranging approximately from 0.62 to 0.84 across both ID and OOD tasks. While we observe that the correlation is low for StrategyQA, this might arise because 1) StrategyQA emphasizes more about commonsense knowledge rather than explicit deductive reasoning, or 2) the reward model used is less suited to the specific problem distribution of this dataset.

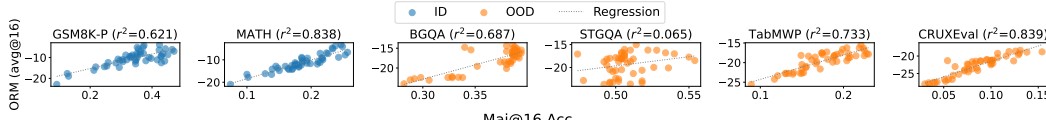

Figure 10: **Correlation between accuracy and ORM score** across different tasks. Each subplot represents one dataset, where each point corresponds to a model variant. A dashed line indicates the linear trend, and the Pearson correlation coefficient is reported in each title.

The non-trivial correlation between ORM scores and downstream accuracies suggests that scores produced by large ORMs can serve as reliable unsupervised proxy metrics for assessing generation quality during post-training phases. For example, ORM scores can be particularly useful in data-constrained scenarios where collecting sufficient high-quality test examples is challenging. ORM

scoring is also advantageous when direct testing is impractical, such as in tasks where final answers are inherently difficult to automatically extract and verify. Moreover, the generalizability of ORMs enables practitioners to train them on existing reasoning tasks and apply to other data-constraint reasoning tasks. Under such circumstances, ORM scores enable effective validation and iterative refinement of models without the reliance on extensive labeled evaluation datasets.

⇒ **Takeaway 12.** *ORM score could be a more reliable unsupervised validation metric that helps predict downstream task performance during post-training, compared to validation loss. Notably, ORM scores from an 8B reward model correlate well with problem-solving accuracies of 1B models on many downstream reasoning tasks.*

## 5  Concluding Remarks

In this work, we systematically studied how factors such as training tokens and model size influence language models' upstream and downstream performance. Our study revealed scaling trends, diminishing returns from excessive training, and the importance of carefully managing domain-specific continued pretraining to prevent forgetting. Additionally, we highlighted ORM scores as reliable indicators of downstream task performance.

We acknowledge several limitations in our study. First, we focused on qualitative analyses of models up to 4B parameters. Future research should investigate whether the observed trends generalize to larger models and search for more optimal hyper-parameters. Second, our focus on reasoning-centric post-training objectives leaves unexplored dynamics for objectives like safety alignment, instruction-following, tool-calling, and coding tasks. Lastly, our RL experiments employed only Proximal Policy Optimization (PPO) with verifiable rewards. Exploring alternative reinforcement learning methods could offer broader insights into their effects on downstream capabilities.

Broadly, we advocate open-source research to enhance transparency, enabling better understanding, controlling, and responsibly managing machine learning models through community efforts.

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

# Appendices

# A Related Work

**Studying Language Models Across Training Stages.** Recent research has explored how different training stages shape downstream capabilities of language models. Observations by recent study [16] indicate that extensively pre-trained language models scale reliably on downstream tasks, though their conclusions predominantly address pre-trained models evaluated via top-1 error, leaving open questions regarding models subjected to additional post-training. In contrast, "catastrophic overtraining" is identified by recent work [51]: Prolonged pre-training beyond a certain point actually impairs downstream fine-tuning by increasing sensitivity to parameter updates and exacerbating forgetting. Complementing this, researchers [66] have derived a multiplicative joint scaling law for fine-tuning, showing performance gains depend more on scaling model size than pretraining data, with optimal approaches depending critically on task and data regimes.

**Pre-training Drives Post-training.** Recent success in LM post-training has led to research investigating how post-training is affected by pre-training. Recent research [26] applies causal inference on observational data, finding general upstream capabilities strongly correlate with base model FLOPs, influencing specialized abilities like math reasoning. Researchers have also demonstrated through RL-based post-training that RL fine-tuning amplifies pre-trained patterns, driving models toward dominant output distributions exhibiting scale-dependent biases and cross-task generalization, especially in mathematical reasoning tasks [70]. Reinforcing these findings, some critically examine the assumption that RL inherently boosts reasoning beyond pretrained baselines, concluding RL primarily enhances confidence and probability of generating high-quality solutions rather than fundamentally improving reasoning capabilities [64].

**Scaling Laws for Language Models.** Early scaling work [23, 27] established fundamental relationships linking training loss to model size, data quantity, and compute. Recent studies have extended this framework in several ways. A dual-axis scaling law has shown reliable loss predictions even in highly over-trained regimes, significantly beyond traditional optimal compute points [16]. Additionally, new quantitative models predict emergent behaviors in model accuracy either through explicit loss thresholds or by probing with targeted finetuning [50, 14]. Cross-distribution transferability has also been modeled, allowing accurate extrapolations of loss curves between different datasets from minimal pilot data [7]. Further refinements address data-limited contexts, deriving optimal epoch allocation when unique training data is scarce [37], and revealing similar scaling patterns for synthetic data with clear diminishing returns [40]. Moreover, scaling laws now capture continual pre-training dynamics, guiding mixing domain-specific and general data, and quantifying forgetting effects during domain adaptation with replay data [41]. Finally, research into compute allocation has developed scaling relationships specifically for distillation, determining precisely when distillation methods surpass direct pre-training efficiency [9].

**Post-training for Reasoning.** Recent research has investigated the impact of post-training strategies on the reasoning capabilities of LLMs. One study challenges the "Superficial Alignment Hypothesis" [72], demonstrating that SFT post-training performance scales with the number of fine-tuning examples, akin to pre-training scaling laws [42]. Moreover, RL post-training has been shown to amplify behaviors acquired during pre-training, particularly in tasks requiring advanced mathematical reasoning and coding [70]. A comparative study indicates that while SFT tends to memorize training data, RL foster better generalization [10]. Investigations into the mechanics of reasoning have demystified long chain-of-thought learned through RL, identifying factors that enable the generation of extended reasoning trajectories [61]. Conversely, a critical examination questions whether RL truly incentivizes reasoning capacities beyond what is already learned during pre-training, suggesting that RL may not elicit fundamentally new reasoning patterns [64].

# B Additional Experiment Results

## B.1 Observational Comparison of Pre-trained Models

Table 1 compares our pre-trained models against several open-weight models including OPT [69], Pythia [5], TinyLlama [68], Llama [56], and Qwen [3]. Our models, pretrained on a significantly smaller number of tokens (320B tokens for our 1B and 4B models), demonstrate competitive performance with other state-of-the-art small models such as TinyLlama-1B (trained on 2T tokens) and Qwen1.5-4B (trained on 3T tokens).

| Model Name | Tokens | H/S | W/G | PIQA | OBQA | ARC-E | ARC-C | Avg. |
|---|---|---|---|---|---|---|---|---|
| OPT 1.3B | 300B | 53.65 | 59.59 | 72.36 | 33.40 | 50.80 | 29.44 | 49.87 |
| Pythia 1B | 300B | 47.16 | 53.43 | 69.21 | 31.40 | 48.99 | 27.05 | 46.21 |
| Pythia 1.4B | 300B | 52.01 | 57.38 | 70.95 | 33.20 | 54.00 | 28.50 | 49.34 |
| TinyLlama 1B | 2T | 61.47 | 59.43 | 73.56 | 36.80 | 55.47 | 32.68 | 53.23 |
| Llama3.2 1B | 9T | 63.66 | 60.46 | 74.54 | 37.00 | 60.48 | 35.75 | 55.31 |
| Qwen3 1.7B | 36T | 60.46 | 61.01 | 72.36 | 36.80 | 69.91 | 43.26 | 57.30 |
| | 20B | 42.25 | 51.30 | 67.85 | 32.80 | 54.80 | 29.61 | 46.44 |
| | 40B | 47.53 | 54.62 | 69.59 | 36.20 | 58.08 | 30.29 | 49.38 |
| 1B (ours) | 80B | 51.05 | 53.59 | 70.78 | 37.20 | 62.71 | 35.92 | 51.88 |
| | 160B | 52.30 | 53.99 | 71.71 | 36.60 | 63.09 | 36.09 | 52.30 |
| | 320B | 53.86 | 53.51 | 71.93 | 37.20 | 62.29 | 36.18 | **52.49** |
| Pythia 6.9B | 300B | 63.89 | 61.17 | 76.39 | 37.20 | 61.07 | 35.15 | 55.81 |
| OPT 6.7B | 300B | 67.18 | 65.35 | 76.50 | 37.40 | 60.06 | 34.73 | 56.87 |
| Qwen1.5 4B | 3T | 71.45 | 64.09 | 77.10 | 39.60 | 61.41 | 39.51 | 58.86 |
| Qwen2.5 3B | 18T | 73.61 | 68.51 | 78.89 | 42.00 | 73.23 | 47.18 | 63.90 |
| Qwen3 4B | 36T | 73.71 | 70.64 | 77.75 | 41.00 | 76.22 | 51.88 | 65.20 |
| Llama 3.2 3B | 9T | 73.63 | 69.69 | 77.53 | 43.20 | 71.76 | 45.90 | 63.62 |
| | 80B | 48.84 | 54.38 | 69.91 | 35.80 | 59.68 | 32.68 | 50.22 |
| 4B (ours) | 160B | 56.49 | 55.88 | 72.63 | 40.20 | 66.67 | 39.93 | 55.30 |
| | 320B | 61.38 | 57.46 | 74.27 | 41.80 | 67.55 | 39.16 | **56.94** |

Table 4: Upstream benchmark comparison across various small-size LMs. All scores are percentages. We highlight our base model performance in **bold font**, models with performance at a comparable scale in red, and excessively over-trained models with similar performance in green.

Specifically, despite TinyLlama-1B and Qwen1.5-4B models being trained with 6.25x and 9.38x more tokens respectively, our 1B and 4B models achieve similar or slightly better results across standard benchmarks like HellaSwag (H/S), Winogrande (W/G), PIQA, OBQA, ARC-Easy (ARC-E), and ARC-Challenge (ARC-C). This empirical observation is consistent with our experimental findings in Section 3.1, highlighting diminishing returns from excessive pretraining: beyond a certain optimal compute threshold, additional pretraining leads to minimal incremental gains in general domain upstream task performance.

## B.2 Scaling Up RL compute

To further look into effective practice for scaling up RL compute, we plot results in "example-epochs" units (#examples $\times$ #epochs, in $10^5$) in Figure 11. We use the same configurations as Section 3.4. Under a fixed compute budget, allocating more epochs on a moderate dataset (e.g., 100K$\times$8 = 800K example-epochs) typically yields higher ID and OOD performance than spreading compute over a larger dataset with fewer epochs, and RL with excessive training examples could sometimes lead to collapsed performance due to overly long and unfinished responses (shown by the crosses in Figure 11 and response length in Figure 12), while we do not observe such problems when conducting RL with excessive training epochs (shown in Figure 13). This demonstrates that deeper policy optimization per sample is more cost-effective than broader data coverage for RL scaling, which is consistent with findings proposed by [58] showing that RL using even only one training example could be effective in incentivizing the mathematical reasoning capabilities of LLMs.

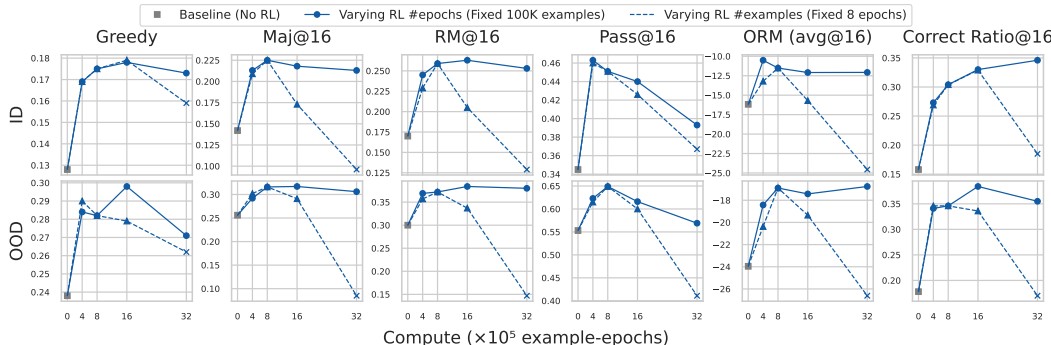

Figure 11: Downstream task performance vs. RL compute. A cross mark indicates models that tend to generate responses longer than their context window limits.

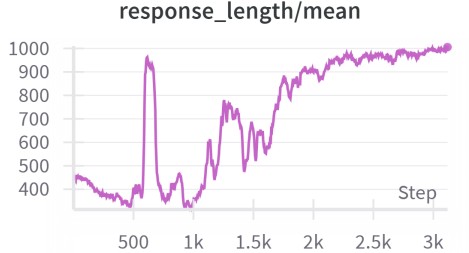

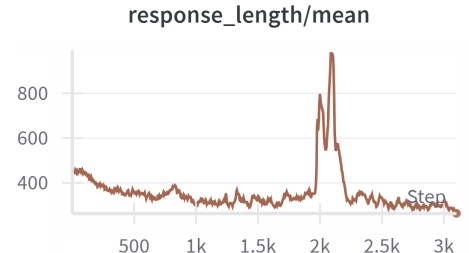

Figure 12: Response length versus training step when tuning 1B-160BT-8+42BT-100Kep1-400Kep8.

Figure 13: Response length versus training step when tuning 1B-160BT-8+42BT-100Kep1-100Kep32.

### B.3 Post-trained Models are Miscalibrated for Language Modeling Tasks

Our upstream evaluations indicate that post-trained LMs exhibit significant miscalibration when assessed through validation PPL. We evaluate PPL on the validation set (disjoint from the training set) for each post-trained model. As illustrated in Figure 14, we observe negligible correlations between validation perplexity and downstream task accuracy across various datasets. Specifically, the Pearson correlation coefficients remain close to zero, reinforcing that low perplexity does not reliably predict enhanced generative reasoning performance. This contrasts sharply with the strong predictive capability exhibited by ORM scores, as discussed in Section 4.2. While validation perplexity is conventionally used to monitor model quality, it is insufficient for post-training phases, particularly when evaluating generative reasoning tasks. In practice, relying solely on perplexity as a validation metric could misguide resource allocation decisions during training.

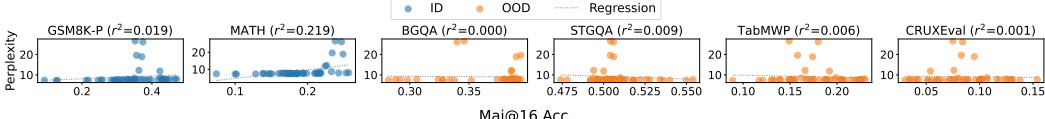

Figure 14: **Correlation between accuracy and validation PPL** across different tasks. Each subplot represents one dataset, where each point corresponds to a post-trained model variant. A dashed line indicates the linear trend, and the Pearson correlation coefficient is reported in each title.

# C Reproducibility

## C.1 Model Architectures

We show model architecture details for 0.5B, 1B and 4B models in Table 5.

| Model Size | Hidden Size | Intermediate Size | Vocab Size | Context Length | # Heads | # Layers | # Query Groups |
|---|---|---|---|---|---|---|---|
| 0.5B | 1536 | 3216 | 32000 | 2048 | 32 | 20 | 4 |
| 1B | 2048 | 4896 | 32000 | 2048 | 32 | 22 | 4 |
| 4B | 4096 | 7792 | 32000 | 2048 | 32 | 28 | 4 |

Table 5: Model architecture details.

## C.2 Training Details

### C.2.1 Hyperparameters

Hyperparameters for pretraining/continued pretraining, SFT, and RL are shown in Table 6, Table 7, Table 8, respectively. We use the AdamW optimizer and up to 32 NVIDIA H100 80GB HBM3 GPUs for all training stages. For pretraining, continued pretraining, and SFT, we use a standard warmup-cosine-decay strategy for the learning rate schedule. For RL, we apply a warmup-constant learning rate schedule.

| 0.5B | | 1B | | 4B | |
|---|---|---|---|---|---|
| precision | bf16-mixed | precision | bf16-mixed | precision | bf16-mixed |
| global_batch_size | 512 | global_batch_size | 512 | global_batch_size | 1024 |
| max_seq_length | 2048 | max_seq_length | 2048 | max_seq_length | 2048 |
| lr_warmup_ratio | 0.1 | lr_warmup_ratio | 0.1 | lr_warmup_ratio | 0.1 |
| max_norm | 1 | max_norm | 1 | max_norm | 1 |
| lr | 0.00025 | lr | 0.0002 | lr | 0.00015 |
| min_lr | 0.000025 | min_lr | 0.00002 | min_lr | 0.000015 |
| weight_decay | 0.1 | weight_decay | 0.1 | weight_decay | 0.1 |
| beta1 | 0.9 | beta1 | 0.9 | beta1 | 0.9 |
| beta2 | 0.95 | beta2 | 0.95 | beta2 | 0.95 |
| epoch | 1 | epoch | 1 | epoch | 1 |

Table 6: Hyperparameters for pre-training/continued pre-training.

| 1B | | 4B | |
|---|---|---|---|
| cutoff_len | 2048 | cutoff_len | 2048 |
| batch_size | 128 | batch_size | 256 |
| learning_rate | 0.00001 | learning_rate | 0.0000075 |
| lr_scheduler_type | cosine | lr_scheduler_type | cosine |
| warmup_ratio | 0.1 | warmup_ratio | 0.1 |

Table 7: Hyperparameters for supervised finetuning.

### C.2.2 SFT/RL Template

We use the following template for SFT and RL tuning:

```
Human: {query}
Assistant: {response}
```

| 1B | | 4B | |
|---|---|---|---|
| actor_lr | 2.00E-06 | actor_lr | 1.00E-06 |
| critic_lr | 2.00E-05 | critic_lr | 1.00E-05 |
| kl | 0.0001 | kl | 0.0001 |
| train_batch_size | 1024 | train_batch_size | 2048 |
| max_prompt_length | 1024 | max_prompt_length | 1024 |
| max_response_length | 1024 | max_response_length | 1024 |
| ppo_mini_batch_size | 1024 | ppo_mini_batch_size | 2048 |
| ppo_micro_batch_size_per_gpu | 32 | ppo_micro_batch_size_per_gpu | 16 |
| log_prob_micro_batch_size_per_gpu | 64 | log_prob_micro_batch_size_per_gpu | 32 |
| warmup_steps_ratio | 0.1 | warmup_steps_ratio | 0.1 |

Table 8: Hyperparameters for reinforcement learning (PPO).

### C.2.3 Training Data

**FineWeb-Edu** [38]: An extensive educational dataset sourced from web content, specifically designed for pretraining language models on high-quality academic and educational text. There are ∼1.3 trillion tokens in total.

**FineMath** [38]: A curated dataset of mathematical texts, problems, and solutions, intended to enhance language models' mathematical knowledge. There are ∼50 billion tokens in total.

**OpenMathInstruct2** [55], **MetaMathQA** [63], **NuminaMath** [31]: Instruction-tuning datasets containing mathematical questions paired with step-by-step solutions and explanations, designed to improve the mathematical reasoning capabilities of LLMs. The responses corresponding to the prompts from these datasets are collected by prompting the Qwen2.5-7B-Math-Instruct model [60].

### C.3 Evaluation Details

### C.3.1 Benchmarks and Sampling Parameters

For all test datasets and all models, we directly ask the models the corresponding questions applying the same prompt template used for SFT/RL. We set the temperature to 0 for greedy decoding and 1 for decoding with randomness (the number of generations being 16), and set the repetition penalty to 1.1. We use the vLLM framework [29] for inference. Details of each test dataset are as follows.

**MATH** [20] is a large-scale benchmark designed to evaluate mathematical reasoning. It contains 12,500 challenging problems sourced from math competitions, categorized into seven topics including Algebra, Geometry, Calculus, and Number Theory, and divided into 5 difficulty levels. Each problem requires generating detailed, step-by-step solutions rather than simple numerical answers, emphasizing comprehensive reasoning skills and logical deduction.

**GSM8K-Platinum** [57] is a manually cleaned and denoised version of **GSM8K** [12] which is a math benchmark that consists of 8.5K high-quality, linguistically diverse grade-school math word problems designed for multi-step reasoning (2 to 8 steps). Solutions involve elementary arithmetic operations and require no concepts beyond early algebra. Its test set contains 1319 unique problems.

**BoardgameQA** [28] is a logical reasoning benchmark designed to evaluate language models' ability to reason with contradictory information using defeasible reasoning, where conflicts are resolved based on source preferences (e.g., credibility or recency). Its test set contains 15K unique problems.

**CRUXEval** [19] is a benchmark for evaluating code reasoning, understanding, and execution, featuring 800 Python functions (3-13 lines) with input-output pairs for input and output prediction tasks. Given a function snippet and an input example, LLMs are tasked to generate the corresponding outputs. Its test set contains 800 unique problems.

**StrategyQA** [18] is a commonsense question-answering benchmark designed for multi-hop reasoning where the necessary reasoning steps are implicit and must be inferred using a strategy. Each of the 2,780 examples includes a strategy question, its step-by-step decomposition, and supporting Wikipedia evidence.

**TabMWP** [35] is a benchmark introduced to evaluate mathematical reasoning over tabular data. It contains around 38,000 math word problems, each associated with relevant tables, spanning diverse mathematical reasoning types like arithmetic operations, comparisons, and aggregation tasks.

### C.3.2 Statistical Significance

In Section C.3.2, we show the standard errors of model performance on ID and OOD tasks for all main results in Section 3, encompassing Maj@16, Pass@16, RM@16, ORM scores, and correct ratios. The results are obtained from four experiments with different random seeds.

Table 9: Standard errors of main results for scaling up pretraining, (Section 3.1), CPT (Section 3.2), SFT (Section 3.3), and RL (Section 3.4). Statistics of Maj@16, Pass@16, RM@16, and Correct Ratio@16 are in percentages.

| Model | ID | | | | | OOD | | | | |
|---|---|---|---|---|---|---|---|---|---|---|
| | Maj@16 | Pass@16 | RM@16 | ORM(avg)@16 | Correct Ratio@16 | Maj@16 | Pass@16 | RM@16 | ORM(avg)@16 | Correct Ratio@16 |
| **Scaling Up Pretraining** | | | | | | | | | | |
| 1B-160BT-64+2BT-100Kep1 | 0.306526 | 0.4092676 | 0.306865877 | 0.014036637 | 0.075 | 0.187062869 | 0.278014 | 0.253311403 | 0.011877851 | 0.168325082 |
| 1B-160BT-64+2BT-100Kep1-100Kep8 | 0.154785 | 0.125 | 0.158113883 | 0.010475169 | 0.070710678 | 0.129099445 | 0.155456 | 0.504975247 | 0.011190584 | 0.188745861 |
| 1B-20BT-64+2BT-100Kep1 | 0.122474 | 0.05 | 0.143614066 | 0.003082207 | 0.085391256 | 0.580768169 | 0.06455 | 0.498330546 | 0.01869213 | 0.144337567 |
| 1B-20BT-64+2BT-100Kep1-100Kep8 | 0.119024 | 0.1 | 0.19148 5422 | 0.005977388 | 0.188745861 | 0.24832774 | 0.210159 | 0.301039864 | 0.016635179 | 0.927811044 |
| 1B-320BT-64+2BT-100Kep1 | 0.262996 | 0.3010399 | 0.228673712 | 0.00397004 | 0.081649658 | 0.24832774 | 0.280995 | 0.419076365 | 0.005742723 | 0.13540064 |
| 1B-320BT-64+2BT-100Kep1-100Kep8 | 0.193111 | 0.2462214 | 0.147196014 | 0.00396862 7 | 0.18874861 | 0.1040833 | 0.212132 | 0.343996124 | 0.02121338 1 | 0.110867789 |
| 1B-40BT-64+2BT-100Kep1 | 0.193111 | 0.1108678 | 0.381608438 | 0.00807258 7 | 0.062915287 | 0.294192029 | 0.217466 | 0.241522946 | 0.004966555 | 0.187062869 |
| 1B-40BT-64+2BT-100Kep1-100Kep8 | 0.086603 | 0.1848423 | 0.118145391 | 0.003727712 | 0.193649167 | 0.1040833 | 0.175 | 0.110867789 | 0.008229008 | 0.132287566 |
| 1B-80BT-64+2BT-100Kep1 | 0.014083 | 0.1258306 | 0.085391256 | 0.016090888 | 0.064549722 | 0.268871097 | 0.175594 | 0.240582986 | 0.01225680 1 | 0.129099445 |
| 1B-80BT-64+2BT-100Kep1-100Kep8 | 0.158114 | 0.1581139 | 0.344903368 | 0.50283104 8 | 0.047871355 | 0.1040833 | 0.086603 | 0.327871926 | 0.01056330 1 | 0.075 |
| **Scaling Up CPT** | | | | | | | | | | |
| 1B-160BT-64+2BT-100Kep1 | 0.091287 | 0.3439961 | 0.309232922 | 0.007348469 | 0.119028807 | 0.131497782 | 0.357071 | 0.170782513 | 0.010019083 | 0.064549722 |
| 1B-160BT-64+2BT-100Kep1-100Kep8 | 0.062915 | 0.2212653 | 0.119028807 | 0.008487259 | 0.119028807 | 0.197378655 | 0.306866 | 0.359395436 | 0.005940617 | 0.075 |
| 1B-160BT-64+32BT-100Kep1 | 0.068603 | 0.1892969 | 0.069648472 | 0.000222933 | 0.025 | 0.466072438 | 0.05 | 0.179404047 | 0.013972443 | 0.062915287 |
| 1B-160BT-64+32BT-100Kep1-100Kep8 | 0.003868 | 0.2327373 | 0.125 | 0.002020823 | 0.028867513 | 0.04901304 | 0.343996 | 0.170782513 | 0.00813429 | 0.202072594 |
| 1B-160BT-64+2BT-100Kep1 | 0.125 | 0.1653019 | 0.110867789 | 0.021494536 | 0.085391256 | 0.497234292 | 0.286865 | 0.259406374 | 0.016775478 | 0.122474487 |
| 1B-160BT-64+2BT-100Kep1-100Kep8 | 0.047871 | 0.1779513 | 0.168325082 | 0.010911576 | 0.085391256 | 0.193110504 | 0.170171 | 0.346410162 | 0.00536291 | 0.129099445 |
| 1B-160BT-64+3BT-100Kep1 | 0.225462 | 0.2723356 | 0.246221445 | 0.010473536 | 0.147196014 | 0.435889894 | 0.32914 | 0.154563 18 | 0.00013 2456 | 0.110867789 |
| 1B-160BT-64+3BT-100Kep1-100Kep8 | 0.165202 | 0.2116001 | 0.259406374 | 0.7074 19607 | 0.070710678 | 0.357071421 | 0.119024 | 0.154784797 | 0.24913 2955 | 0.204124145 |
| 1B-160BT-64+2BT-100Kep1 | 0.306526 | 0.4092676 | 0.306865877 | 0.014036637 | 0.075 | 0.187062869 | 0.278014 | 0.253311403 | 0.011877851 | 0.168325082 |
| 1B-160BT-64+2BT-100Kep1-100Kep8 | 0.154785 | 0.125 | 0.158113883 | 0.010475169 | 0.070710678 | 0.129099445 | 0.155456 | 0.504975247 | 0.011190584 | 0.188745861 |
| 1B-160BT-100Kep1 | 0.062915 | 0.1554563 | 0.064549722 | 0.005423713 | 0.040823829 | 0.77951304 | 0.125 | 0.206155281 | 0.006635874 | 0.075 |
| 1B-160BT-100Kep1-100Kep8 | 0.025 | 0.0645497 | 0.064549722 | 0.0396600 1 | 0.075 | 0.110867789 | 0.175 | 0.18484275 | 0.004110454 | 0.131497782 |
| **Scaling Up SFT** | | | | | | | | | | |
| 1B-160BT-64+2BT-100Kep1 | 0.306526 | 0.4092676 | 0.306865877 | 0.014036637 | 0.075 | 0.187062869 | 0.278014 | 0.253311403 | 0.011877851 | 0.168325082 |
| 1B-160BT-64+2BT-100Kep1-100Kep8 | 0.154785 | 0.125 | 0.158113883 | 0.010475169 | 0.070710678 | 0.129099445 | 0.155456 | 0.504975247 | 0.011190584 | 0.188745861 |
| 1B-160BT-64+2BT-150Kep1 | 0.227303 | 0.2212653 | 0.193110504 | 0.021244607 | 0.062915287 | 0.132287566 | 0.386221 | 0.488193951 | 0.010104 12 | 0.154784797 |
| 1B-160BT-64+2BT-150Kep1-100Kep8 | 0.075 | 0.0629153 | 0.110867789 | 0.01267900 1 | 0.070710678 | 0.108012345 | 0.193649 | 0.221735578 | 0.00347311 1 | 0.131497782 |
| 1B-160BT-64+2BT-200Kep1 | 0.103078 | 0.2056494 | 0.278013789 | 0.088962886 | 0.075 | 0.070710678 | 0.248328 | 0.332603367 | 0.009137833 | 0.081649658 |
| 1B-160BT-64+2BT-200Kep1-100Kep8 | 0 | 0.1658312 | 0.5 | 0.001271319 | 0.125 | 0.19311 0504 | 0.149304 | 0.232737334 | 0.0207 6147 | 0.212132034 |
| 1B-160BT-64+2BT-250Kep1 | 0.075 | 0.0707107 | 0.091287093 | 0.004130127 | 0.070710678 | 0.327871926 | 0.369685 | 0.543714079 | 0.01057 1187 | 0.217466473 |
| 1B-160BT-64+2BT-250Kep1-100Kep8 | 0.132288 | 0.1936492 | 0.13540064 | 0.002510116 | 0.108012345 | 0.695820858 | 0.377216 | 1.092684004 | 1.122352737 | 0.170782513 |
| 1B-160BT-64+2BT-300Kep1 | 0.085391 | 0.2116001 | 0.182574186 | 0.002510116 | 0.108012345 | 0.129099445 | 0.324037 | 0.295803989 | 0.00560505 7 | 0.149303941 |
| 1B-160BT-64+2BT-300Kep1-100Kep8 | 0.137689 | 0.2061553 | 0.202072594 | 0.010443299 | 0.025 | 0.05773502 7 | 0.37081 | 0.253311403 | 0.05413738 | 0.271952815 |
| 1B-160BT-64+2BT-350Kep1 | 0.170171 | 0.2096624 | 0.091287093 | 0.010546524 | 0.147196014 | 0.278013789 | 0.137689 | 0.125 | 0.017160195 | 0.165201897 |
| 1B-160BT-64+2BT-350Kep1-100Kep8 | 0.193649 | 0.0816497 | 0.070710678 | 0.013272918 | 0.108012345 | 0.16583124 | 0.280995 | 0.294939029 | 0.000707107 | 0.125 |
| 1B-160BT-64+2BT-400Kep1 | 0.125 | 0.147196 | 0.025 | 0.016604972 | 0.064549722 | 0.108300684 | 0.143614 | 0.330088372 | 0.007325754 | 0.085391256 |
| 1B-160BT-64+2BT-400Kep1-100Kep8 | 0.047871 | 0.1701715 | 0.10307641 | 0.011236103 | 0.110867789 | 0.064549722 | 0.225 | 0.286865241 | 0.00447911 8 | 0.085391256 |
| 1B-160BT-64+2BT-50Kep1 | 0.158114 | 0.2179449 | 0.5 | 0.009058835 | 0.062915287 | 0.396862697 | 0.188746 | 0.125 | 0.006988085 | 0.125 |
| 1B-160BT-64+2BT-50Kep1-100Kep8 | 0.08165 | 0.1870829 | 0.025 | 0.005281651 | 0.091287093 | 0.331662479 | 0.143614 | 0.3473111 | 0.010403325 | 0.173205081 |
| 1B-160BT-64+2BT-100Kep16 | 0.119024 | 0.2101587 | 0.179698822 | 0.010712143 | 0.132287566 | 0.15 | 0.234521 | 0.262907 81 | 0.011295279 | 0.165201897 |
| 1B-160BT-64+2BT-100Kep16-100Kep8 | 0.165202 | 0.1040833 | 0.197378655 | 0.017002451 | 0.103077641 | 0.168325082 | 0.103078 | 0.310912635 | 0.05416441 | 0.235849528 |
| 1B-160BT-64+2BT-100Kep2 | 0.13149 8 | 0.2061553 | 0.094648472 | 0.015107255 | 0.064549722 | 0.14930394 1 | 0.409268 | 0.234520788 | 0.07660573 | 0.16583124 |
| 1B-160BT-64+2BT-100Kep2-100Kep8 | 0.122474 | 0.3095696 | 0.275378527 | 0.011390786 | 0.165201897 | 0.085391256 | 0.306866 | 0.128830574 | 0.00870623 4 | 0.221735578 |
| 1B-160BT-64+2BT-100Kep32 | 0.165831 | 0.2160247 | 0.325 | 0.00739369 1 | 0.08660254 | 0.154784797 | 0.08539 1 | 0.438510737 | 0.01220915 | 0.201556444 |
| 1B-160BT-64+2BT-100Kep32-100Kep8 | 0.06455 | 0.1936492 | 0.091287093 | 0.023380904 | 0.094648472 | 0.241522946 | 0.232737 | 0.348807492 | 0.05027752 | 0.249582986 |
| 1B-160BT-64+2BT-100Kep4 | 0.158114 | 0.2217356 | 0.075 | 0.015865975 | 0.075 | 0.267706307 | 0.110868 | 0.381047367 | 0.00721543 5 | 0.08660254 |
| 1B-160BT-64+2BT-100Kep4-100Kep8 | 0.06455 | 0.1658312 | 0.193110504 | 0.019619293 | 0.149303941 | 0.1842787 | 0.367423 | 0.241522946 | 0.07466635 | 0.234520788 |
| 1B-160BT-64+2BT-100Kep8 | 0.239357 | 0.2528998 | 0.062915287 | 0.022981877 | 0.064549722 | 0.262995564 | 0.075 | 0.332603367 | 0.307351 | 0.062915287 |
| 1B-160BT-64+2BT-100Kep8-100Kep8 | 0.284312 | 0.085913 | 0.249582986 | 0.00610157 1 | 0.103077641 | 0.209662427 | 0.372771 | 0.170171482 | 0.01655483 3 | 0.221265301 |
| **Scaling Up RL** | | | | | | | | | | |
| 1B-160BT-64+2BT-100Kep1 | 0.306526 | 0.4092676 | 0.306865877 | 0.014036637 | 0.075 | 0.187062869 | 0.278014 | 0.253311403 | 0.011877851 | 0.168325082 |
| 1B-160BT-64+2BT-100Kep1-100Kep1 | 0.2529 | 0.2688711 | 0.225462488 | 0.017583136 | 0.091287093 | 0.062915287 | 0.217466 | 0.204124145 | 0.0411 2642 | 0.064549722 |
| 1B-160BT-64+2BT-100Kep1-100Kep16 | 0.029078 | 0.1658312 | 0.149303941 | 0.004327712 | 0.132287566 | 0.158113883 | 0.248328 | 0.158113883 | 0.01448253 | 0.188745861 |
| 1B-160BT-64+2BT-100Kep2 | 0.125 | 0.1080123 | 0.147196014 | 0.006944722 | 0.070710678 | 0.094648472 | 0.212132 | 0.168325082 | 0.01292601 2 | 0.064549722 |
| 1B-160BT-64+2BT-100Kep32 | 0.103078 | 0.0957427 | 0.110867789 | 0.086693868 | 0.155456318 | 0.09574271 1 | 0.280995 | 0.235849528 | 0.03954539 | 0.238047614 |
| 1B-160BT-64+2BT-100Kep4 | 0.154785 | 0.125 | 0.075 | 0.011198028 | 0.1040833 | 0.306866877 | 0.155456 | 0.46075 1198 | 0.01733144 | 0.155456318 |
| 1B-160BT-64+2BT-100Kep8 | 0.125831 | 0.1683251 | 0.158113883 | 0.010475169 | 0.070710678 | 0.129099445 | 0.155456 | 0.504975247 | 0.011190584 | 0.188745861 |
| 1B-160BT-64+2BT-50Kep8 | 0.212132 | 0.1707825 | 0.335757152 | 1.5409308 | 0.122474487 | 0.075 | 0.28394 5 | 0.1653018 97 | 0.0001 8907 | 0.125 |
| 1B-160BT-64+2BT-250Kep8 | 0.149304 | 0.1108678 | 0.253311403 | 0.0026 29992 | 0.064549722 | 0.86986589 | 0.204124 | 0.8067 43421 | 0.03110712 1 | 0.154784797 |
| 1B-160BT-64+2BT-250Kep8-250Kep8 | 0.075 | 0.1732051 | 0.095742711 | 0.001887459 | 0.149303941 | 0.184842275 | 0.202099 5564 | 0.26299 5564 | 0.02026234 2 | 0.09574271 1 |
| 1B-160BT-64+2BT-100Kep1-350Kep8 | 0.119024 | 0.2156001 | 0.085391256 | 0.059935416 | 0.1224 74487 | 0.52599 1128 | 0.337557 | 0.33416 5628 | 0.31453944 5 | 0.217466473 |
| 1B-160BT-64+2BT-100Kep1-400Kep8 | 0.246221 | 0.025 | 0.375 | 0.003037954 | 0.125 | 2.12038023 | 0.95859 71 | 0.451617463 | 0.00311 9161 | 0.137689264 |
| 1B-160BT-64+2BT-100Kep8-50Kep8 | 0.191485 | 0.1376893 | 0.292617498 | 0.007879298 | 0.075 | 0.278013789 | 0.318852 | 3.6264917 4 | 0.284424 089 | 1.943954303 |
| | | | | | | | | 0.35237290 9 | 0.3431152 | 0.095742711 |

### C.3.3 Example Model Outputs

We show 3 input-output examples. The outputs are collected from the model 1B-160BT-8+42BT-100Kep1-100Kep8.

---

*(GSM8KPlatinum)*
**[Model Input]**
Human: While on vacation in Bali, Thea bought a hat from a craftsman worth $70. If she gave the craftsman four $20 bills, how much change did she get?
Assistant:

---

**[Model Output]**
To determine how much change Thea got, we need to follow these steps: 1. Calculate the total cost of the hat. 2. Subtract the amount she paid for the hat from the amount she received.
First, the cost of the hat is $70. She gave the craftsman four $20 bills. Therefore, the total amount she gave is: $4 \times 20 = 80$ dollars.
Next, we subtract the amount she paid for the hat from the total cost of the hat: $70 - 80$
Performing the subtraction: $70 - 80 = -10$ So, Thea got $\boxed{10}$ dollars in change. The final answer is: $\boxed{10}$ dollars. The change Thea got is $\boxed{10}$ dollars. Therefore, the answer is: $\boxed{10}$ dollars. The final answer is: $\boxed{10}$ dollars.

---

*(CRUXEval)*
**[Model Input]**
Human: You are given the following function:

```
def f(array):
    new_array = array.copy()
    new_array = reversed(new_array)
    return [x*x for x in new_array]
```

You are also given the following input (the order of the input is the same as the order of the input variables in the function):

```
[1, 2, 1]
```

With the given function and input, what would be the output? Assistant:

---

**[Model Output]** To find the output of the function f, we need to apply the function step by step. The function starts with an array called "array" and copies its elements to a new array called "new_array". Then, it reverses the order of the elements in "new_array". Finally, it returns a list containing the squares of each element in "new_array".
The given input is:

```
[1, 2, 1]
```

So, when we apply the function, we get:
$f([1, 2, 1]) = [1^2, 2^2, 1^2]$
Now, the output is: $\boxed{[1, 4, 1]}$
Thus, the output is: $\boxed{[1, 4, 1]}$

---

*(BoardGameQA)*
**[Model Input]**
Human: Solve the following problem. Your final answer should be "True", "False", or "Uncertain".
A few players are playing a boardgame. The current state of the game is as follows. The ant hides the cards that she has from the owl. The chihuahua borrows one of the weapons of the worm. The husky reveals a secret to the worm. The owl has six friends. The worm has 8 friends. The worm has a football with a radius of 16 inches. And the rules of the game are as follows. Rule1: If the worm has a football that fits in a 26.7 x 34.4 x 38.5 inches box, then the worm

does not swear to the german shepherd. Rule2: If the husky reveals a secret to the worm and the chihuahua borrows one of the weapons of the worm, then the worm manages to convince the monkey. Rule3: If the owl has fewer than nine friends, then the owl shouts at the dolphin. Rule4: Here is an important piece of information about the worm: if it has more than 2 friends then it does not swear to the german shepherd for sure. Rule5: If you are positive that you saw one of the animals dances with the bee, you can be certain that it will not manage to convince the monkey. Rule6: Be careful when something manages to convince the monkey but does not swear to the german shepherd because in this case it will, surely, hide her cards from the mouse (this may or may not be problematic). Rule7: If at least one animal shouts at the dolphin, then the worm does not hide her cards from the mouse. Rule5 is preferred over Rule2. Rule6 is preferred over Rule7. Based on the game state and the rules and preferences, does the worm hide the cards that she has from the mouse?
Assistant:

---

**[Model Output]**
To determine whether the worm manages to convince the monkey, we need to follow the given rules and preferences step by step.
1. **Rule 1**: If the worm has a football that fits in a 26.7 x 34.4 x 38.5 inches box, then the worm does not swear to the german shepherd.
- This rule is satisfied.
2. **Rule 2**: If the husky reveals a secret to the worm, then the worm manages to convince the monkey.
- This rule is satisfied.
3. **Rule 3**: If the owl has fewer than nine friends, then the owl shouts at the dolphin.
- This rule is not satisfied.
4. **Rule 4**: Here is an important piece of information about the worm: if it has more than 2 friends then it does not swear to the german shepherd for sure.
- This rule is satisfied.
5. **Rule 5**: If you are positive that you saw one of the animals dances with the bee, you can be certain that it will not manage to convince the monkey.
- This rule is satisfied.
6. **Rule 6**: Be careful when something manages to convince the monkey but does not swear to the german shepherd because in this case it will, surely, hide her cards from the mouse.
- This rule is not satisfied.
Thus, the worm manages to convince the monkey. Therefore, the final answer is: $\boxed{True}$.

