# OpenReview forum: "EvoLM: In Search of Lost Training Dynamics for Language Model Reasoning"
_NeurIPS.cc/2025/Conference — NeurIPS 2025 oral_

### Official Review · Reviewer_3Qki · 2025-06-28

**Clarity:** 3
**Significance:** 3
**Originality:** 2
**Rating:** 5
**Confidence:** 4

**Summary:**

This paper releases a suite of over 100 language models with varying decisions across the full training pipeline: from pretraining (#tokens, #params in {0.5B, 1B, 4B}) to continued pretraining (#tokens new mix, #tokens original mix), SFT (#examples, #epochs), RL (#examples, #epochs). Data is fixed for pretraining (FineWeb-EDU), continued pretraining (FineMath), SFT (Various math data), and RL (same sources as SFT). Alignment algorithm (PPO) is also fixed, and a fixed set of hyperparams are used for each model size. They provide analysis of how changes in the varied decisions impact evaluations upstream (Cloze Form HellaSwag, Winogrande, PIQA, OBQA, ARC) and downstream (generative GSM8K + MATH, CruxEval, BGQA, TabMWP, StrategyWA). These analyses are used to provide 13 practical takeaways for model developers, such as not over-pretraining and the need to replay original pretraining data during the continued pretraining.

**Questions:**

- What can the authors say about how easy it will be to build on their suite? Whats it like to run post training steps on top of one of your pretrained models? If you want to run a new pretraining experiment but with a smaller configuration how easy is that? How about availability of intermediate checkpoints? Optimizer states? How easy is it to run different kinds of evaluations?


- Would love to see an analysis of the relationship of upstream to downstream performance (and how this may be altered by various training decisions)


- Font in figure in figures is frequently too small to read.

- line103 define ORM score before use

- how do I interpret ORM avg@16 in figure 2. why are the scores negative? Is higher better? Maybe include a hyperlink in the caption to section 2.2 for metric definitions.

- table 1 is difficult to read, could be better as a figure. at least tell us the take away in the caption.

- Table 2 doesn’t say what param size and pretraining tokens trained are used in these experiments.

- maybe put the value into the x axis label not just the caption, e.g. figure 5 should have x axis “SFT Epochs”

- what are the x marks in the fig 7 b? also it would be interesting to see this subplot but with more than 1 epoch since it seems that epoching multiple times is very important based on figure 7 a.

- I like the use of color in fig 8! Maybe also just quickly say SFT allocation increases further to the right.

- It would be great to have a table or figure telling us what are all the models released.

- table 3 is hard to parse but I’m not sure how to explain it better since it’s subtle, could this be a line plot instead with the 20BT x position highlighted?

- Figure 9 is waaay too small, maybe just say the R^2 values in a table or make it big enough to see the plots.

- 32K vocab size seems like a bit of an uncommon choice (I think of >64K being more common). You could argue this is well motivated for helping you explore smaller, less expressive models but regardless might be worth pointing out somewhere outside of the appendix.

- What is the LR schedule for pretraining? Cosine?

- How were the hyperparams selected?

- A possibility (but totally fine if out of scope): the analysis in fig 3 could also compare using a moving weight space averaging of checkpoints or some other form of regularization instead of data replay. Also a lack of annealing of the LR of the base model before CPT may play a role in disruption caused by the data distribution shift (but I think you drop the LR pretty low right?) But maybe your LR is too high during CPT or your warm up is too short?

- line 205: Does overly large epochs mean too many epochs or that each epoch is too large (i.e., too many examples)?

- figure 7: don’t use shades of blue to indicate different metrics as its hard to distinguish them. Use differences in hue and or changes in marker.

- is the correlation in 4.2 between ORM and Downstream performance both on fully post trained models or is the ORM score on just the mid trained model before it’s post-trained?

**Ethical Concerns:**

["NO or VERY MINOR ethics concerns only"]

**Final Justification:**

I think this suite of models will be quite impactful, as I am not aware of previous sweeps over both pretraining and post training configurations in a single suite. I'm increasing my score since I can see in the table of experiments provided in the author response that the experiment sweeps are denser than I realized. I also think the findings in the paper are helpful bring together evidence for a wide set of best practices in pre and post training all in one set of experiments. My other concerns are mostly just reasonable limitations that the authors embrace as well motivated scoping decisions (e.g. focusing on math evals since they're more verifiable). I recommend this paper be accepted.

**Limitations:**

yes

**Quality:**

3

**Strengths And Weaknesses:**

strengths:

To my knowledge this is the first work that releases a suite sweeping over token/param ratios in pretraining and also conducts post training over may configurations. The most expensive to reproduce result open sourced by this suite is confirming the effect of over pretraining an reducing SFT and SFT + RL performance. This was previously observed by Springer et al (2025), as cited in this paper, but they only examined intermediate checkpoints rather than doing runs with different scheduled endings. Most of the rest of the analyses in the paper just build off the 1B-160BT model and thus could have just been conducted on top of an off the shelf base model. But future work can use their artifacts to explore the effect of token/param ratio on numerous other aspects of the post-training pipeline.

Beyond the compute expense of trying multiple pretraining configurations, this suite of artifacts will also save developer time simply by having saved checkpoints for lots of different post training configurations all in one place. This is not as expensive to reproduce, but if this suite is well organized it could speed up research built on top of it. In particular I see a great role for this suite in pedagogy. It would be great tool for students learning about the full language model training pipeline.

This work also examines various subtleties of post-training: the relative importance of CPT and SFT (possibly hinting at CPT cannibalizing SFT?), The effect of RL on confidence, trade offs between SFT and RL for ID and OOD performance.

weakness:

While still an interesting avenue of inquiry, this suite has a domain specific mid and post training focus on math, contrary to the more common practice of post training models for generalist performance. The evaluation does includes OOD eval but even these are very targeted tasks and do not include general evaluations of preference alignment eg MT-bench, alpaca-eval.

The current paper and appendix don’t clearly list everything that will be available in the release. One can assume that hopefully all the models mentioned analyses will be released, but if these are all that will be released it’s worth noting that then the suite sweeps much more sparsely over some options than others (for example the 4B models participate in far fewer of the analyses than the 1B models). It’s understandable that a full grid search over all the complex configurations would be very challenging to put together, but given the expense that goes into pretraining I hope the cheaper to sweep stages of training are more densely sampled.

Much of the practical utility of this study consists of finding specific good settings for things like how much data to replay in CPT, but it’s not discussed much how one should go about generalizing these findings to other configurations (e.g. bigger models). The answer may just be that without sampling more than one point over model size for example, we can’t do better than hoping that the ratio of replay to new CPT data is constant, which is fine. But some discussion of what we can know and how we might know more would be important to have.

---

> ### Author Rebuttal · Authors · 2025-07-31
>
> # Response to Reviewer 3Qki
>
> ## General issues:
>
> > W1: While still an interesting avenue of inquiry, ...
>
> It is correct that we intentionally focuses on the domain of mathematics, but we consider this to be a deliberate methodological choice rather than a limitation. The common practice of training for generalist performance is often evaluated using subjective, preference-based benchmarks like MT-Bench, which are noisy, difficult to reproduce, and can obscure the true effects of specific training decisions. By anchoring the entire study in a domain with verifiable, objective correctness, we can isolate and measure the fundamental dynamics of training—such as saturation points and the trade-offs between SFT and RL with a scientific rigor that would be impossible in a purely subjective setting.
>
> > W2: The current paper and appendix don’t clearly list everything that will be available in the release...
>
> Due to anonymization policies, we cannot release assets at this stage. However, all models and code and data discussed in the paper will be released post-acceptance, along with full training and evaluation documentations. While the 4B sweep is sparser due to resource constraints, we prioritized high-resolution sweeps at the 1B scale and plan to expand post-training variations for 4B after acceptance.
>
> > W3: Much of the practical utility of this study consists of finding specific good settings for things like how much data to replay in CPT, ...
>
> Thank you for raising a crucial point regarding the generalizability of specific hyperparameters, but the primary contribution of our paper is not to prescribe universal constants, but rather to identify and validate the fundamental dynamics at play, thereby providing the very methodology for "how we might know more." For example, by demonstrating that a modest replay budget of around 5% effectively mitigates catastrophic forgetting (Takeaway 4), the study provides an invaluable, empirically-grounded starting point that drastically reduces the search space for researchers working with different model sizes or domains.
>
> Our paper's value lies in establishing what to look for and why it matters, which equips researchers with the knowledge that they must consider a data replay strategy and should begin their search in a low-single-digit percentage range, rather than having to discover this principle from scratch. Thus, the work offers not a static answer, but a validated, efficient template for inquiry.
>
> > Q1: What can the authors say about how easy it will be to build on their suite? ...
>
> We expect that the practical usability of the model suite is important, and the project was designed with this exact concern in mind. The release accompanying our paper goes beyond merely providing model weights; it includes the entire open-source training, post-training, and evaluation pipeline used in the study.
>
> This was a deliberate design decision we made to ensure the suite is a modular and extensible toolkit, not a static artifact. Consequently, running post-training steps on our models is as simple as loading a chosen pre-trained checkpoint into the provided pipeline; conducting a new, smaller-scale pre-training experiment involves modifying the transparently documented configurations (Appendix C); and running different evaluations means plugging a new dataset into our open-source evaluation harness.
>
> By providing not just the models but the complete, end-to-end code, we have deliberately lowered the barrier to entry for precisely the kinds of follow-on research and custom experiments you describe. For example, we expect our checkpoints are useful for downstream performance scaling law fits and other reliability studies like data attribution.
>
> > Q2: Would love to see an analysis of the relationship of upstream to downstream performance ...
>
> Thank you for the suggestion. We agree that understanding the relationship between upstream and downstream performance is interesting, and in fact, our study has incorporated investigations into this aspect across training stages. Specifically:
>
> 1. Section 3.1 shows that while more extensive pretraining consistently improves upstream performance (e.g., lower perplexity), downstream performance saturates and can even degrade—highlighting a clear decoupling between upstream and downstream gains in the overtraining regime.
> 2. Section 3.2 further demonstrates that continued pretraining (CPT) on domain-specific data leads to degraded upstream performance due to catastrophic forgetting, yet this step is critical for improving downstream task performance, especially for in-domain reasoning tasks.
>
> These results reveal that training decisions such as scaling pretraining or incorporating replay in CPT can alter the alignment—or misalignment—between upstream and downstream behavior. We will clarify this connection more explicitly in the next version to highlight that this relationship is a core focus of our analysis.
>
> ## Clarification issues:
>
> > how do I interpret ORM avg@16 in figure 2...
>
> Thank you for pointing this out. ORM avg@16 refers to the average ORM score over 16 sampled generations per input prompt. As explained in Section 2.2, this scalar score—produced by an external reward model—serves as a proxy for solution quality, where **higher scores indicate better responses**. The scores can be negative because the ORM model outputs unnormalized scores centered around zero, depending on the alignment and quality of the response relative to the input. To avoid confusion, in our next draft we will (1) explicitly state in the figure caption that higher is better, (2) clarify that negative values are possible due to the ORM’s scoring convention, and (3) include a hyperlink to Section 2.2 for detailed metric definitions.
>
> > Table 2 doesn’t say what param size and pretraining tokens trained are used in these experiments.
>
> Thank you for pointing this out. In line 155, we have pointed out that we use 1B model pretrained with 160 billion tokens. We will add this information to the caption of the table.
>
> > what are the x marks in the fig 7 b? ...
>
> We apologize for the confusion. The x-marks in Figure 7b denote models that frequently generate overly long responses exceeding the context window, which leads to truncated outputs and degraded performance—an issue we refer to as model collapse. These failures typically arise when training with very large RL datasets (e.g., ≥350K examples) for one epoch, where the model over-optimizes toward verbosity rather than correctness. We agree that comparing against multi-epoch RL runs—as in Figure 7a—would be insightful, since multiple epochs appear to yield more stable and robust improvements without triggering collapse. We will consider including such a comparison in the revision or supplement to illustrate how scaling depth (epochs) versus breadth (examples) influences model behavior under RL.
>
> > 32K vocab size seems like a bit of an uncommon choice (I think of >64K being more common)...
>
> Thank you for the thoughtful observation. You're right—the 32K vocabulary size is smaller than what is commonly used in larger-scale models (often >64K). We intentionally chose 32K to align with our focus on studying smaller models (1B and 4B) under controlled compute budgets, where a more compact vocabulary helps reduce embedding size and training overhead. We agree this design choice deserves more visibility and will add a brief explanation in the main text to clarify its motivation beyond what's currently noted in the appendix.
>
>
> > What is the LR schedule for pretraining? Cosine?
>
> Yes, we use cosine LR scheduler.
>
>
> > How were the hyperparams selected?
>
> Thank you for the question. We used a consistent and carefully tuned set of hyperparameters across all experiments to isolate the effects of scaling compute, model size, and training stages. For pretraining, we adopted a cosine learning rate schedule with warmup, and tuned the base learning rate and batch size separately for 1B and 4B models to ensure stable convergence and optimal upstream task performance. For post-training stages (SFT, RL), we selected hyperparameters based on preliminary grid searches, guided by prior work on hyperparameter scaling and training stability in LLMs, including Wortsman et al. [1] and Zhang et al. [2]. To improve transparency, we will add a table of all key hyperparameters to the Appendix in the newer version, including learning rates, batch sizes, optimizers, and scheduling details for each training phase.
>
> [1] Small-scale proxies for large-scale transformer training instabilities
>
> [2] How does critical batch size scale in pre-training?
>
> > A possibility (but totally fine if out of scope)...
>
> Thank you for the thoughtful suggestions. You're absolutely right that techniques such as weight space averaging or additional forms of regularization during CPT could help mitigate forgetting, and we agree that comparing those approaches to data replay would be valuable. Similarly, adjusting the LR schedule could influence the stability of adaptation. However, we consider these directions beyond the scope of our current study, which focuses on data-centric strategies like replay for mitigating distribution shift. We do use a decayed learning rate at the end of pretraining and adopt conservative learning rate settings for CPT, but a deeper sweep over schedule configurations remains an interesting avenue for future work.
>
> > line 205: Does overly large epochs mean too many epochs or that each epoch is too large (i.e., too many examples)?
>
> It means "too many epochs".
>
> > is the correlation in 4.2 between ORM and Downstream performance both on fully post trained models or is the ORM score on just the mid trained model before it’s post-trained?
>
> They are fully post-trained models.
>
> ## Formatting issues:
>
> Per formatting issues you have mentioned, we will modify each of them in our next version. Thank you very much for pointing them out!

---

> > ### Comment · Reviewer_3Qki · 2025-08-06
> > **Thanks for response**
> >
> > I appreciate the authors' response. I'm particularly happy to hear they plan to release all their artifacts and are invested the task of making the suite user friendly and well maintained.
> >
> > Per weakness 2, I would love to see a table or diagram that enumerates all of the training experiments (Param counts, number of tokens, configurations etc) that will be released. This will make it easier to understand what parts of the experiment sweeps are denser or more sparse. Is it possible to specify all the experiments run as a set of ranges over configuration options? I understand that this may be more complex than a simple grid, as there are branches of experiments where, say, only some pretraining experiments get a certain set of post training conditions. But anything you can do to try and get that information all in one place would be very helpful.

---

> ### Author Response · Authors · 2025-08-08
> **Respond to Reviewer 3Qki**
>
> Thank you for your response! In our newest draft, we will include a full table listing all models we will release. Due to limited space here, we show a dense format of the table:
>
> ---
>
> | Model Size |   Pre-train Tokens (BT)  | CPT (BT) (FW/FM)                                               | SFT (epochs × examples)               | RL (epochs × examples)                |
> | :--------: | :----------------------: | :------------------------------------------------------------- | :------------------------------------ | :------------------------------------ |
> |    0.5B    | 10, 20, 40, 80, 160, 320 | None                                                           | None                                  | None                                  |
> |     1B     |   20, 40, 80, 160, 320   | FW8+FM2/12/22/32/42, FW1.6+FM48.4, FW16+FM34, FM10/20/30/40/50 | ep1/2/4/8/16/32 × 100k/200k/300k/400k | ep1/2/4/8/16/32 × 100k/200k/300k/400k |
> |     4B     |       80, 160, 320       | FW8+FM2/12/22/32/42                                            | ep1/2/4/8/16/32 × 100k/200k/300k/400k | ep1/2/4/8/16/32 × 100k/200k/300k/400k |
>
>
> ---
>
>
> - Pre-train tokens (BT): BT = billion tokens (e.g., 1B-160BT = 1B params, 160B tokens)
>
> - CPT: Continued Pretraining on combinations of FineWeb-Edu (FW) and FineMath (FM) with various splits, e.g., FW8+FM42 = 8B tokens FineWeb + 42B tokens FineMath
>
> - SFT: Supervised fine-tuning on up to 32 epochs and 400k examples (combinations of ep1, ep2, ep4, ep8, ep16, ep32 × 100k, 200k, 300k, 400k)
>
> - RL: Reinforcement learning on the same SFT axes
>
>
> We keep maintaining the models and will incorporate denser sweep over the configurations. For example, we will include 2B models in our newest model suite.

---

### Official Review · Reviewer_mn1m · 2025-07-03

**Clarity:** 3
**Significance:** 4
**Originality:** 2
**Rating:** 5
**Confidence:** 4

**Summary:**

This paper presents a highly systematic and transparent empirical study of large language model (LLM) training dynamics across all major phases: pre-training, continued pre-training, supervised fine-tuning (SFT), and reinforcement learning (RL). The authors train over 100 decoder-only LLaMA-2 models with 1B and 4B parameters from scratch, controlling for variables such as model size, token budgets, and data mixture. The study evaluates both upstream (language modeling) and downstream (reasoning/problem-solving) performance on a suite of in-domain and out-of-domain tasks. Key findings include: (1) diminishing returns and even potential performance degradation with excessive pre-training, (2) the critical role of domain-specific continued pre-training with replay to mitigate catastrophic forgetting, (3) the interplay and trade-offs between SFT and RL for in-domain and OOD generalization, and (4) practical recipes and heuristics for effective training at each stage. All models, datasets, and code are released for reproducibility and further research.

**Questions:**

1. **Architectural Generalization:**
   *To what extent do you expect your key empirical findings (such as the pre-training saturation points, replay ratios, and RL/SFT trade-offs) to hold for other architectures (e.g., Mistral, GPT-2/3, Mixture-of-Experts, or even non-transformer LMs)? Could the authors comment on whether they have tried, or plan to try, these recipes with alternative backbones? This is a crucial point for the impact and universality of the work.*

2. **Deeper Analysis of Negative Transfer:**
   *The observed degradation of downstream performance with excessive pre-training is both surprising and important. Can the authors provide further analysis, ablation, or speculative hypotheses on the underlying mechanism? Is it optimization (e.g., overfitting, sharpness), distributional mismatch, or something else? Any additional insight here would elevate the paper's significance.*

3. **Broader Task Diversity:**
   *How might the best practices and trade-offs identified here change for other core LLM tasks—especially dialogue, tool use, multi-turn instruction, code synthesis, or safety-critical objectives? Have the authors considered expanding the evaluation suite or at least discussing potential limitations in generalizing these findings?*

4. **Computational Cost and Reproducibility at Scale:**
   *While the open release is commendable, running such a suite remains computationally expensive. Are there any practical recommendations or heuristics for researchers with more limited resources, or plans to provide smaller-scale ablations for the community?*

Score could be increased if the authors provide empirical evidence or at least in-depth analysis that the key findings are robust across architectures, or offer new insight into the mechanisms underlying negative transfer from over-pretraining.

**Ethical Concerns:**

["NO or VERY MINOR ethics concerns only"]

**Final Justification:**

The authors' response mostly addressed my questions and comments, but did not provide sufficient new information for me to raise my score. I'll maintain my score of 5 and vote for accept.

**Limitations:**

Yes, the authors have a dedicated discussion of limitations and are appropriately upfront about the primary weaknesses of their study. One additional limitation, however, is the architectural specificity: all findings are from LLaMA-2-based models, and this may restrict the universality of their "best practices" until further replication.

**Paper Formatting Concerns:**

No.

**Quality:**

3

**Strengths And Weaknesses:**

## Strengths and Weaknesses

### Strengths

- **Extremely systematic and transparent:** The study is one of the most comprehensive empirical investigations of LLM training dynamics to date. The careful control of experimental variables, end-to-end training from scratch, and consistent evaluation protocol enable clear, actionable insights.
- **Open science:** Release of 100+ model checkpoints, all training data, and full code pipeline. This sets a strong standard for reproducibility and enables the community to build on this work.
- **Practical best practices:** The paper distills an array of "recipes" and trade-offs for LLM practitioners, offering guidance on how much to scale each stage and how to balance domain adaptation, generalization, and forgetting.
- **Novel empirical insights:** The finding that excessive pre-training can degrade downstream performance after SFT/RL is notable, as is the systematic exploration of replay ratios for catastrophic forgetting.
- **Clear writing and presentation:** The structure, figures, and explanations are clear and facilitate understanding, even for complex, multi-stage training dynamics.

### Weaknesses

- **Architectural specificity:** All experiments are conducted on the LLaMA-2 architecture. This is a major limitation, as some conclusions and observed trends (e.g., the precise saturation regimes, the degree of negative transfer, RL/SFT trade-offs) may not generalize to other architectures (e.g., Mistral, Mamba, Mixture-of-Experts, or non-transformer architectures). Without replication on other backbones, it's unclear how universal the best practices are.
- **Interpretability of negative transfer:** While the paper robustly documents the phenomenon of diminishing returns and even performance degradation from over-pretraining, it stops short of providing a deep analysis or theoretical explanation. Is this due to optimization issues, overfitting, pre-training/fine-tuning distribution mismatch, or something more subtle? Some ablation or hypothesis testing in this direction would increase the work's significance.
- **Empirical, not theoretical:** The paper is almost entirely empirical. While this is not a flaw per se, the lack of any attempt to connect to theoretical frameworks (e.g., statistical learning theory, information theory, optimization landscapes) somewhat limits the generalizability and scientific depth of the conclusions. However, this is a minor issue compared to the main contributions.

---

> ### Author Rebuttal · Authors · 2025-07-31
>
> # Response to Reviewer mn1m
>
> Thank you for your thoughtful and constructive feedback. Below, we address your comments and suggestions point by point:
>
> ## About architectural specificity
> > W1: Architectural specificity: All experiments are conducted on the LLaMA-2 architecture. This is a major limitation, as some conclusions and observed trends (e.g., the precise saturation regimes, the degree of negative transfer, RL/SFT trade-offs) may not generalize to other architectures (e.g., Mistral, Mamba, Mixture-of-Experts, or non-transformer architectures). Without replication on other backbones, it's unclear how universal the best practices are.
> >
> > Q1: Architectural Generalization: To what extent do you expect your key empirical findings (such as the pre-training saturation points, replay ratios, and RL/SFT trade-offs) to hold for other architectures (e.g., Mistral, GPT-2/3, Mixture-of-Experts, or even non-transformer LMs)? Could the authors comment on whether they have tried, or plan to try, these recipes with alternative backbones? This is a crucial point for the impact and universality of the work.
>
> The qualitative trends—such as pre-training saturation, the need for data replay in CPT to combat forgetting, and the SFT/RL trade-off between in-domain specialization and out-of-domain generalization—are expected to hold for other dense, decoder-only transformers like Mistral or GPT, as evidence by many related works in the literature [1, 2]. These are fundamental dynamics of learning from data, not just architectural quirks. However, the exact quantitative numbers, like the 80x-160x saturation point or the 5% replay ratio, would almost certainly need recalibrating for different models. For radically different architectures like Mixture-of-Experts, these recipes may require significant adaptation, and confirming this generalizability is a promising direction for future work.
>
> [1] D-CPT Law: Domain-specific Continual Pre-Training Scaling Law for Large Language Models
>
> [2] Overtrained Language Models Are Harder to Fine-Tune
>
> ## About negative transfer
> > W2: Interpretability of negative transfer: While the paper robustly documents the phenomenon of diminishing returns and even performance degradation from over-pretraining, it stops short of providing a deep analysis or theoretical explanation. Is this due to optimization issues, overfitting, pre-training/fine-tuning distribution mismatch, or something more subtle? Some ablation or hypothesis testing in this direction would increase the work's significance.
> >
> > Q2: Deeper Analysis of Negative Transfer: The observed degradation of downstream performance with excessive pre-training is both surprising and important. Can the authors provide further analysis, ablation, or speculative hypotheses on the underlying mechanism? Is it optimization (e.g., overfitting, sharpness), distributional mismatch, or something else? Any additional insight here would elevate the paper's significance.
>
> Thank you for highlighting the importance of analyzing the mechanism behind the degradation from excessive pretraining. While our current work focuses only on empirical study of this effect across stages of the training pipeline, we believe our results are highly consistent with—and complemented by—recent theoretical insights. Springer et al. [1] provide a compelling explanation via the concept of progressive sensitivity, which increases with the amount of pretraining. Their analysis shows that as pretraining progresses, model parameters become more fragile to post-training updates. This increased sensitivity causes greater forgetting or distortion of acquired knowledge during post-training, leading to what they call catastrophic overtraining. They demonstrate that this degradation is inevitable when pretraining is extended without constraints and is exacerbated by distribution mismatch between pretraining and post-training tasks.
>
> In our work, we extended the study to both upstream and downstream tasks and in-domain and out-of-domain tasks, and observe similar phenomenon. Models trained beyond a certain token threshold begin to show degraded performance on downstream reasoning tasks, even though upstream perplexity continues to improve. This aligns with their theoretical finding that the inflection point—the point beyond which sensitivity dominates the benefits of additional pretraining—marks the onset of degradation. We view a more formal exploration of the sensitivity–plasticity trade-off as an exciting future direction, and we appreciate the reviewer’s encouragement to deepen this analysis.
>
> [1] Overtrained Language Models Are Harder to Fine-Tune
>
> ## Other questions
> > W3: Empirical, not theoretical: The paper is almost entirely empirical. While this is not a flaw per se, the lack of any attempt to connect to theoretical frameworks (e.g., statistical learning theory, information theory, optimization landscapes) somewhat limits the generalizability and scientific depth of the conclusions. However, this is a minor issue compared to the main contributions.
>
> We acknowledge that our study is primarily empirical by design, as our goal was to provide a rigorous, controlled, and transparent investigation into training dynamics across the full lifecycle of language models. That said, we agree that connecting our findings to broader theoretical frameworks could enrich the scientific depth and interpretability of our results. We view this as a valuable direction for future work and hope that our released models and pipeline can serve as a foundation for such theoretical explorations by the community.
>
> > Q3: Broader Task Diversity: How might the best practices and trade-offs identified here change for other core LLM tasks—especially dialogue, tool use, multi-turn instruction, code synthesis, or safety-critical objectives? Have the authors considered expanding the evaluation suite or at least discussing potential limitations in generalizing these findings?
>
> While it is true that our empirical results of are focused on reasoning tasks, this was a deliberate methodological choice necessary to establish a clear, reproducible baseline. As we acknowledge in limitations, the study's focus leaves dynamics for tasks like tool use and safety unexplored. However, the work's primary contribution is not a universal recipe but a foundational framework for analysis. We argue that the core principles identified—such as pre-training saturation, catastrophic forgetting during CPT, and the trade-off between specialization (SFT) and generalization (RL) are fundamental learning dynamics that will likely persist across different objectives. The key change will be in the nature of the trade-offs: for dialogue, the SFT/RL balance might shift from logical correctness to one between stylistic alignment and helpfulness; for tool use, RL could be rewarded on successful API execution; and for safety, the entire paradigm might be augmented with additional constraint-based methods. Therefore, this study does not claim to have the final answer for all tasks, but rather provides the essential blueprint and a set of validated hypotheses required to begin investigating these more complex and subjective domains systematically.
>
> > Q4: Computational Cost and Reproducibility at Scale: While the open release is commendable, running such a suite remains computationally expensive. Are there any practical recommendations or heuristics for researchers with more limited resources, or plans to provide smaller-scale ablations for the community?
>
> While the computational cost of this suite is indeed significant, its primary purpose is to serve as a foundational, pre-computed roadmap precisely because such large-scale studies are beyond the reach of most researchers. Rather than requiring a full reproduction, the key takeaways from our study are intended to be the heuristics themselves. For instance, our findings on pre-training saturation provide a clear guideline on when to stop costly training runs (Takeaway 2), and the strong correlation between ORM scores and task performance (Takeaway 13) offers a much cheaper proxy for model evaluation than running extensive downstream tests. Therefore, the value for the resource-constrained researcher lies not in re-running our experiments, but in using these validated principles and proxy metrics to make more informed, cost-effective decisions in their own work.

---

### Official Review · Reviewer_nh5Y · 2025-07-03

**Clarity:** 3
**Significance:** 3
**Originality:** 3
**Rating:** 4
**Confidence:** 3

**Summary:**

The paper studies the impact of pretraining data selection on large language model (LLM) performance. It introduces multiple data filtering strategies and evaluates their effectiveness across a wide range of upstream and downstream tasks, model scales, and post-training phases including continued pretraining (CPT), supervised fine-tuning (SFT), and reinforcement learning (RL). The goal is to identify data characteristics that most strongly influence model performance under various resource constraints.

**Questions:**

- The paper examines scaling of CPT, SFT, and RL individually. While this may be outside the current scope, it would be helpful to hear the authors’ thoughts on budget allocation across PT, CPT, SFT, and RL when total compute is fixed.
- In Section 3.3 (Figures 5 and 6), why does scaling the dataset size appear to have more impact than scaling the number of epochs? Any insights into this effect?
- In Section 4.1 (lines 264–273), clarification is needed: does it mean the difference of using intermediate checkpoint is no learning rate decay? Do the authors use EMA checkpoints, and if so, how do they influence performance? Also, any insights on "continue pretraining" to include more tokens (different from CPT)?
- Are the filtering strategies equally effective across all model scales? Some trends (e.g., impact of data entropy) could vary depending on model size, would be good to confirm.
- Do filtering strategies benefit certain task types more than others (e.g., reasoning vs. factual recall)?

**Ethical Concerns:**

["NO or VERY MINOR ethics concerns only"]

**Final Justification:**

I thank the authors for their rebuttal, I will keep my score and stay positive.

**Limitations:**

Limitations are discussed in the paper. No societal concerns are noted.

**Paper Formatting Concerns:**

no formatting concern

**Quality:**

3

**Strengths And Weaknesses:**

## Strength
- The paper is well written, clearly structured, and easy to follow.
- The experiments are comprehensive and systematic, covering multiple model sizes and training phases.
- The direction is practically important: understanding how to best select or filter training data can directly benefit LLM training efficiency and downstream performance.
- The authors’ analysis offers useful insights for practitioners interested in optimizing compute/data tradeoffs for pretraining and post-training.

## Weakness
I am not an expert in this specific subfield, but I find the contribution solid overall. I don't see any real weakness, a minor presentation nit: the use of “—” is inconsistent in lines 268–269 and 278–279. Please see some clarification questions in question section below.

---

> ### Author Rebuttal · Authors · 2025-07-31
>
> # Response to Reviewer nh5Y
>
> We are grateful to the reviewer for the thoughtful and insightful comments to make our paper better. Below, we address each raised point individually.
>
> > W1: I am not an expert in this specific subfield, but I find the contribution solid overall. I don't see any real weakness, a minor presentation nit: the use of “—” is inconsistent in lines 268–269 and 278–279. Please see some clarification questions in question section below.
>
> Thank you for your advice. We will revise the paper accordingly in the next version.
>
> > Q1: The paper examines scaling of CPT, SFT, and RL individually. While this may be outside the current scope, it would be helpful to hear the authors’ thoughts on budget allocation across PT, CPT, SFT, and RL when total compute is fixed.
>
> Thank you for your thoughtful suggestion. Allocating a fixed compute budget requires a sequential, trade-off-aware approach. First, we should allocate enough pre-training compute to reach the point of diminishing returns—around 80x to 160x the model's size in tokens—but no more, to conserve resources. Next, a critical portion of the budget must go to CPT to adapt the model to a specific domain, incorporating a small amount of data replay (e.g., 5%) to mitigate catastrophic forgetting. Finally, the remaining post-training budget should be split between SFT and RL according to this goal: allocate more to SFT for peak in-domain performance, or more to RL to enhance out-of-domain generalization. While we agree this is an important and practical direction for future work, a comprehensive study of optimal budget allocation is beyond the scope of the current paper.
>
> > Q2: In Section 3.3 (Figures 5 and 6), why does scaling the dataset size appear to have more impact than scaling the number of epochs? Any insights into this effect?
>
> Scaling the number of epochs involves repeatedly showing the model the same fixed dataset, which leads to over-specialization; the model memorizes the specific training examples, causing in-domain performance to saturate and OOD performance to degrade as it fails to learn robust, transferable principles.
> In contrast, scaling the dataset size exposes the model to a wider variety of unique problems, forcing it to learn more general, underlying logic rather than superficial statistical patterns. This leads to more durable improvements for in-domain tasks and better, more stable performance on OOD tasks because the model is genuinely learning problem-solving skills, not just recognizing solutions.
>
> > Q3: In Section 4.1 (lines 264–273), clarification is needed: does it mean the difference of using intermediate checkpoint is no learning rate decay? Do the authors use EMA checkpoints, and if so, how do they influence performance? Also, any insights on "continue pretraining" to include more tokens (different from CPT)?
>
> Yes, the primary difference is that intermediate checkpoints are extracted before full training convergence and learning rate decay has occurred.
>
> We did not use EMA checkpoints in our study. All evaluations are conducted on final checkpoints obtained after full training with complete learning rate schedules.
>
> We did not "continue pretraining" to include more tokens. In our study, we simulate the real-world scenario where practitioners get pretrained checkpoints and start CPT on top of those checkpoints. Moreover, our CPT strategy is data- and objective-specific: we add domain-specific data while interleaving a small amount of replayed general-domain pretraining data. This is different from simply continuing generic pretraining to include more tokens, which could lead to catastrophic forgetting or degraded generalization as shown in Table 2 and Figure 3 (page 5) that effective CPT requires careful data balancing.
>
> > Q4: Are the filtering strategies equally effective across all model scales? Some trends (e.g., impact of data entropy) could vary depending on model size, would be good to confirm.
>
> While we do not explicitly evaluate filtering methods in this study, we designed our experiments to reveal how data scale interact with model scale. For instance, Section 3.1 show that smaller models (e.g., 1B) can outperform larger models (e.g., 4B) under constrained pretraining budgets, suggesting that data efficiency—an implicit effect of filtering—is especially crucial for smaller models. Additionally, in our CPT experiments, the effectiveness of modest replay appears consistent across model sizes, indicating that filtering-like strategies may transfer well across scales. That said, a more systematic study of filtering efficacy at different scales would be a valuable future direction.
>
> > Q5: Do filtering strategies benefit certain task types more than others (e.g., reasoning vs. factual recall)?
>
> Although we do not apply explicit filtering strategies, our results suggest that data curation strategies, e.g., replay or fine-tuning dataset quality control, affect tasks differently. For example, domain-specific CPT yields large improvements on reasoning-heavy tasks like math, whereas excessive pretraining or fine-tuning can sometimes degrade performance on OOD tasks that rely more on general knowledge, e.g., StrategyQA. This points to a possibility that filtering strategies may indeed have differential benefits, e.g. potentially favoring structured reasoning tasks more than open-domain factual recall. We view this as another promising direction for deeper follow-up work.

---

### Official Review · Reviewer_D5SP · 2025-07-11

**Clarity:** 4
**Significance:** 3
**Originality:** 3
**Rating:** 5
**Confidence:** 4

**Summary:**

This paper performs a systematic study of LLM training dynamics across 4 common phases of training: pretraining, continued pretraining, supervised finetuning, and reinforcement learning. To target improvments on math benchmarks, the authors start with pretraining on FineWeb-Edu, perform CPT with FineMath, and finally use a dataset of QA pairs combined from MetaMathQA, OpenMathInstruct2, and NuminaMath for SFT and RL. The paper is organized as a series of takeaways from scaling up each phase of training, and the resulting models from the experiments will be released to the research community.

**Questions:**

1. For the CPT phase, did you start with an annealed checkpoint and warm-up the learning rate or did you just change the dataset during the original anneal.

2. Did you explore using ORM scores from a smaller reward model to predict downstream reasoning tasks? It would be interesting to see the scale of reward model required for your observed correlation to hold.

**Ethical Concerns:**

["NO or VERY MINOR ethics concerns only"]

**Final Justification:**

The paper is clear and has a well-motivated experimental design to explore important interactions between different phases of training. The reproducibility of the paper has been improved by including the details of training hyperparameter. Overall, I recommend acceptance.

**Limitations:**

Yes, the authors have adequately discussed the limitations, primarily the scale at which the experiments are conducted.

**Quality:**

4

**Strengths And Weaknesses:**

**Strengths**

The paper is clear and focused with well-motivated experimental design. The setup of the paper enables the authors to explore important interactions between different phases of training.

From the scaling general-domain pretraining, the authors observe that upstream performance saturates around a token duration of 80x - 160x model size; in addition, by looking at how SFT performs on top of models at different durations, they observe that improvements from domain-specific post-training also tend to stop after this saturation point. Continued pretraining is found to boost SFT and RL finetuning, but a small replay budget was needed to avoid catastrophic forgetting. For both SFT and RL, the authors found points where improvements saturated. Finally, the authors provided some evidence that outcome reward model scores could be used to predict performance on downstream tasks.

Overall, while the observations generally match how LLM training is currently done in practice, the paper adds a methodical and open study of LLM training dynamics at the 1B and 4B scale.

**Weaknesses**

The paper does not have a discussion of how the hyperparameters (like learning rate) were chosen and if the learning rate was swept for any of the post-training phases. Could the authors provide details on this?

---

> ### Author Rebuttal · Authors · 2025-07-31
>
> # Response to Reviewer D5SP
>
> We thank the reviewer for the thoughtful and insightful comments that make our paper better. Below, we address each raised point individually.
>
> > W1: The paper does not have a discussion of how the hyperparameters (like learning rate) were chosen and if the learning rate was swept for any of the post-training phases. Could the authors provide details on this?
>
> Thank you for the question. We used a consistent and carefully tuned set of hyperparameters across all experiments to isolate the effects of scaling compute, model size, and training stages. For pretraining, we adopted a cosine learning rate schedule with warmup, and tuned the base learning rate and batch size separately for 1B and 4B models to ensure stable convergence and optimal upstream task performance. For post-training stages (SFT, RL), we selected hyperparameters based on preliminary grid searches, guided by prior work on hyperparameter scaling and training stability in LLMs, including Wortsman et al. [1] and Zhang et al. [2]. To improve transparency, we will add a table of all key hyperparameters to the Appendix in the newer version, including learning rates, batch sizes, optimizers, and scheduling details for each training phase.
>
> [1] Small-scale proxies for large-scale transformer training instabilities
>
> [2] How does critical batch size scale in pre-training?
>
> > Q1: For the CPT phase, did you start with an annealed checkpoint and warm-up the learning rate or did you just change the dataset during the original anneal.
>
> Thanks for the question. We start with an annealed pretraining checkpoint and warmup the learning rate again.
>
> > Q2: Did you explore using ORM scores from a smaller reward model to predict downstream reasoning tasks? It would be interesting to see the scale of reward model required for your observed correlation to hold.
>
> We did not explore using ORM scores from smaller reward models in this study, as we focused on a strong 8B reward model to ensure high signal quality when correlating with downstream task accuracy. However, we agree that understanding how the scale of the reward model affects its predictive power is an important and promising direction. Future work could investigate whether smaller ORMs can still reliably approximate downstream quality, and whether certain tasks require different levels of capacity in the reward model.

---

> > ### Comment · Reviewer_D5SP · 2025-08-06
> > **Response to Rebuttal**
> >
> > I appreciate the authors’ response to my questions and the addition of a table covering the hyperparameters used for training. Having read all the reviews and responses, I continue to recommend acceptance.

---

### Decision · Program_Chairs · 2025-09-17

**Decision:**

Accept (oral)

**Comment:**

Reviewers are aligned that this is a careful, end-to-end empirical study of LM training dynamics across PT --> CPT --> SFT --> RL, with unusually systematic sweeps (100+ models at 0.5B/1B/4B) with clear takeaways on pretraining saturation, the need for small replay during CPT to mitigate forgetting and SFT/RL trade-offs for ID vs. OOD. The rebuttal helpfully clarified hyperparameter choices and schedules, CPT warm-up, and provided a denser inventory of planned artifacts. Remaining concerns are scope rather than flaws. Overall, the work meets the bar on correctness, transparency and community value.